# Multi-modal Situated Reasoning in 3D Scenes

**Xiongkun Linghu**[1,*], **Jiangyong Huang**[1,2,*], **Xuesong Niu**[1,*], **Xiaojian Ma**[1],
**Baoxiong Jia**[1,†], **Siyuan Huang**[1,†]

[1]State Key Laboratory of General Artificial Intelligence, BIGAI
[2]Peking University

https://msr3d.github.io

## Abstract

Situation awareness is essential for understanding and reasoning about 3D scenes in embodied AI agents. However, existing datasets and benchmarks for situated understanding are limited in data modality, diversity, scale, and task scope. To address these limitations, we propose Multi-modal Situated Question Answering (MSQA), a large-scale multi-modal situated reasoning dataset, scalably collected leveraging 3D scene graphs and vision-language models (VLMs) across a diverse range of real-world 3D scenes. MSQA includes 251K situated question-answering pairs across 9 distinct question categories, covering complex scenarios within 3D scenes. We introduce a novel interleaved multi-modal input setting in our benchmark to provide text, image, and point cloud for situation and question description, resolving ambiguity in previous single-modality convention (*e.g.*, text). Additionally, we devise the Multi-modal Situated Next-step Navigation (MSNN) benchmark to evaluate models' situated reasoning for navigation. Comprehensive evaluations on MSQA and MSNN highlight the limitations of existing vision-language models and underscore the importance of handling multi-modal interleaved inputs and situation modeling. Experiments on data scaling and cross-domain transfer further demonstrate the efficacy of leveraging MSQA as a pre-training dataset for developing more powerful situated reasoning models.

## 1 Introduction

Understanding and interacting with the 3D physical world is fundamental to the development of embodied AI. A central challenge in equipping agents with these capabilities is the integration of situational awareness into models. This is particularly critical given the pivotal role of situation awareness in bridging agents' multi-modal local context (*e.g.*, text descriptions, images, point clouds, *etc*.) with the global environment status, thereby facilitating reasoning and planning in 3D scenes.

However, compared to recent advancement in 3D vision-language learning [9, 1, 72, 24, 30], the study of situation modeling in 3D scenes remained largely underexplored. This is primarily due to the absence of a scalable method to collect diverse multi-modal situational data. Previous studies have mainly relied on simulated environments [63, 55, 47] to generate egocentric observations of virtual agents. These approaches severely limit the quality of situational data due to the constrained diversity and complexity of available synthetic scenes. Recent efforts such as SQA3D [41] have aimed to extend situated understanding to real-world scenes like ScanNet [17] by collecting situated question-answer pairs under imaginative situations represented by locations and orientations in 3D scenes. Nonetheless, this data collection process is prohibitively expensive, thereby restricting the scale of situational data available for model learning and evaluation.

---

*Equal contribution. †Corresponding author.

38th Conference on Neural Information Processing Systems (NeurIPS 2024) Track on Datasets and Benchmarks.

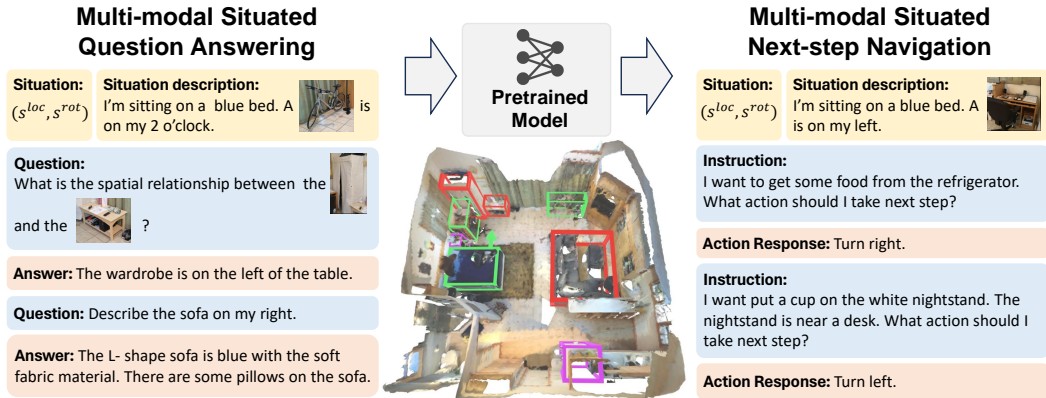

Figure 1: **An overview of benchmarking tasks in MSQA**. We use green boxes for objects mentioned in situation descriptions, red for objects in questions, and purple for objects in navigation instructions.

To address the aforementioned data limitations, we propose, **M**ulti-modal **S**ituated **Q**uestion **A**nswering (MSQA), *a high-quality, large-scale multi-modal dataset for 3D situated reasoning*. Specifically, we develop an automated pipeline for efficient and scalable data collection. First, we generate diverse situations (*i.e.*, spatial locations and viewpoints) in complex real-world scenes sourced from ScanNet [17], 3RScan [60], and ARKitScenes [7]. By adjusting the provided 3D scene graph of each scene based on sampled viewpoints, we create situated scene graphs and use them to generate high-quality situated question-answer pairs by meticulously designing prompts for large language models (LLMs). With this pipeline, we collect 251K situated QA pairs, surpassing existing datasets in scale, question scope, and quality. Additionally, we enrich this dataset with question-answer pairs targeting navigation actions necessary to move between different situations, providing comprehensive learning and evaluation data for embodied navigation. This curated navigation data directly evaluates the transfer of situation understanding from reasoning to action, thereby extending MSQA to cover the full spectrum of embodied tasks in 3D scenes.

With MSQA, we introduce evaluation benchmarks to precisely assess models' situation awareness, addressing limitations of existing benchmarks. Current benchmarks predominantly rely on single-modal descriptions of situations (*i.e.*, texts), which can lead to ambiguity in situation identification, thereby restricting models' situation understanding capability (as shown in Fig. 2). To overcome this, we propose an *interleaved* input setting that integrates *textual descriptions, images, and scene point clouds* to describe situations and questions. This approach resolves ambiguity in situation descriptions and provides a versatile format for broader downstream applications. Leveraging this multi-modal interleaved setting, we establish two challenging benchmarking tasks, Multi-modal Situated Question Answering (MSQA) and Multi-modal Next-step Navigation (MSNN), aimed at evaluating models' capabilities in embodied reasoning and navigation. MSQA expands the scope of existing situated question-answering tasks to encompass object existence, counting, attributes, spatial relationships, *etc*. MSNN simplifies traditional multi-step embodied navigation to a single-step setting, focusing on the immediate next action based on the current situation and navigation target. This design separates long-horizon planning from situated understanding, targeting models' ability to ground actions and transition between actions. We provide an overview of these tasks in Fig. 1.

We provide comprehensive experimental analyses of existing vision-language models on these tasks, exposing their limitations in effectively modeling complex situations and fully utilizing interleaved multi-modal input. In response to identified limitations, we propose a powerful baseline model, MSR3D, specifically designed for handling interleaved multi-modal input with situation modeling that achieves superior results on both MSQA and MSNN. Our additional experiments on data scaling and cross-domain transfer further demonstrate the efficacy of pre-training on our MSQA and showcase the potential of MSR3D. In summary, our key contributions are as follows:

- We introduce MSQA, a large-scale 3D situated reasoning dataset comprising 251K situated QA pairs, curated using a scalable automated data generation pipeline across diverse real-world scenes.
- We propose the use of interleaved multi-modal input setting for model learning and evaluation, establishing two comprehensive benchmarking tasks, MSQA and MSNN, to assess models' capability in situated reasoning and navigation in 3D scenes.

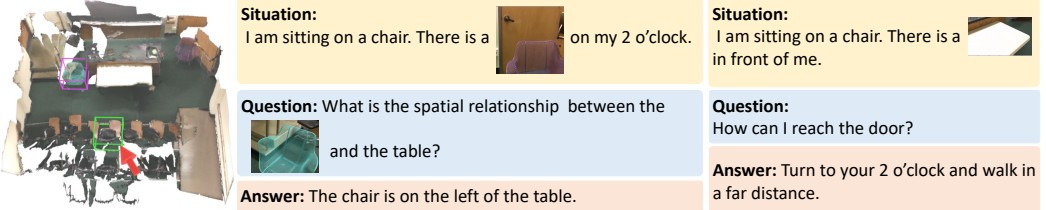

Figure 2: **An illustration on resolving ambiguity with interleaved multi-modal input**. With both chairs highlighted in purple and green boxes having the same textual description "chair is next to the table", one can easily identify the target chair from the candidates by providing an image describing its location.

- We conduct comprehensive experimental analyses comparing existing models with our proposed baseline MSR3D on MSQA and MSNN. We highlight the importance of handling multi-modal interleaved inputs and situation modeling. Through data scaling and cross-domain transfer experiments, we demonstrate the effectiveness of pre-training on MSQA data and the potential of MSR3D for multi-modal situated reasoning in 3D scenes.

## 2 Related Work

**Situated understanding in 3D scenes.** Existing efforts in 3D VL research primarily focus on understanding and reasoning within 3D scenes, including object grounding [9, 1, 70, 11, 46, 32, 64, 67], captioning [14, 12], and question answering [6, 41, 24]. Recently some initiatives propose unified frameworks for various 3D VL tasks [8, 15, 72, 26, 28, 13, 73, 62], yielding promising outcomes. Nonetheless, a prevailing limitation pertains to the absence of situated understanding in these tasks [40, 9, 1, 72, 6, 28], which accounts for a notable gap between 3D VL and embodied AI [4, 54, 49, 52, 55, 2, 20, 65]. While earlier works on situated reasoning [18, 21, 56] typically encompass answering simple questions via exploring the simulative environments, SQA3D [41] introduces real-world scenes with a particular focus on spatial reasoning and scene understanding. SIG3D [43] underscores situational awareness and proposes an effective method to address the challenge. In this paper, we extend the 3D situated reasoning task to more diverse and complex scenarios. Furthermore, we devise innovative multi-modal situated next-step navigation to consolidate the evaluation of situated reasoning.

**LLM-assisted data generation.** Large Language Models (LLMs) exhibit remarkable proficiency in text generation and serve as a valuable resource for collecting diverse textual instruction-following data [61, 57, 16] and multi-modal instruction-following data [38, 34, 37]. This method also exhibits notable promise to aid the scarcity of 3D VL data [40, 28, 35, 30]. However, the quality of LLM-generated data has been a common concern in the community, especially considering the inherent complexity of 3D scenes. To address this problem, existing efforts [24, 50, 28, 35] have improved the LLM prompting techniques and post-processing procedures to enhance the reliability and diversity of LLM-generated data. And some prior works [10, 19] attempt to evaluate the quality of LLM-generated data yet have not resolved the concerns on the quality of LLM-generated data and how it compares to human-annotated data. In this paper, in addition to advanced prompting techniques and post-processing procedures, we also conduct a human study on the quality of LLM-generated data to demonstrate the efficacy of our LLM-assisted data generation approach.

**Interleaved multi-modal understanding.** It has been a critical challenge to precisely delineate the situation within intricate 3D scenes. Natural as it is, adopting textual descriptions [56, 41] may encounter issues of object referral ambiguity, especially when situated within cluttered environments. On the other hand, ego-view visual observations [18, 4, 54, 23, 22] are widely adopted in embodied tasks but bridging the modality gap demands extra training. Recently, interleaved multi-modal data has become prevalent in both VL [58, 3, 27, 34, 71, 29, 69, 35] and embodied AI [53, 31, 36]. In the context of 3D situated reasoning, the interleaved multi-modal format can remedy the ambiguity and thus stands as a general scheme to delineate the situation. Such an interleaved multi-modal scheme strengthens the challenge of our situation reasoning task, requiring comprehensive capabilities of VL grounding and multi-modal situated reasoning.

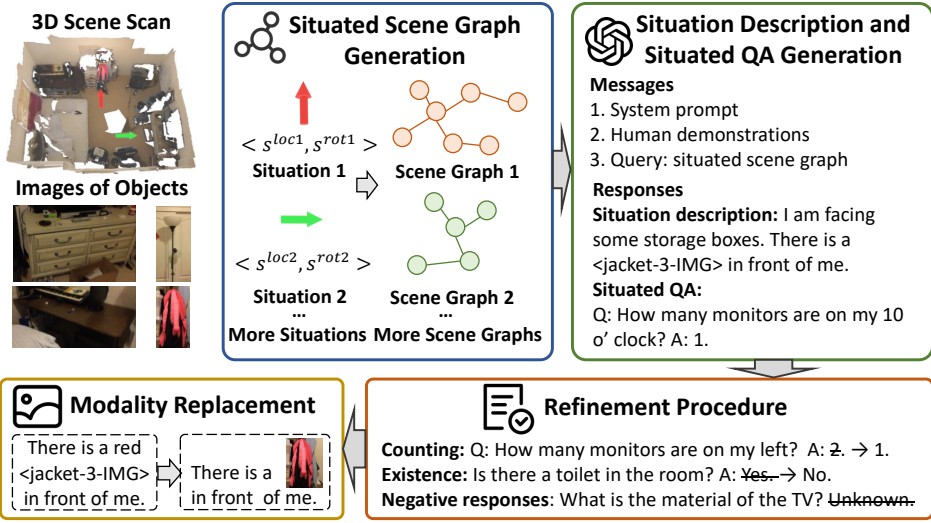

Figure 3: **An overview of our data collection pipeline**, including situated scene graph generation, situated QA pairs generation, and various post-processing procedures.

## 3 Multi-modal Situated Reasoning Dataset

We propose a novel and scalable approach to collecting high-quality 3D situated reasoning data, guided by three core principles: (1) ensuring comprehensive and diverse situations, (2) crafting highly situation-dependent questions with accurate answers, and (3) accommodating the multi-modal interleaved input format for avoiding ambiguities. We construct the MSQA dataset by employing our data collection pipeline on complex real-world scenes sourced from ScanNet [17], 3RScan [60] and ARKitScenes [7]. MSQA comprises 251K multi-modal situated question-answering data. Each data instance can be formulated as a tuple $(\mathbf{P}, \mathbf{S}, \mathbf{q}, \mathbf{a})$, where $\mathbf{P}$ denotes the scene point cloud; $\mathbf{S} = (s^{txt,img}, s^{loc}, s^{rot})$ includes a multi-modal situation description $s^{txt,img}$, the corresponding location $s^{loc}$ and orientation $s^{rot}$, the interleaved multi-modal question $\mathbf{q} = \mathbf{q}^{txt,img}$ collected under situation $\mathbf{S}$, and the ground truth answer $\mathbf{a}$. In the following sections, we delineate our data collection pipeline in Appendix A.2 and present data statistics in Appendix A.6.

### 3.1 Data Collection

As illustrated in Fig. 3, we meticulously devise an LLM-based automatic data collection pipeline comprising three stages: situation sampling, QA pairs generation, and data refinement. Our goal for data collection is to ensure the high quality of generated data. We detail the pipeline below.

**Situation sampling**    The situation consists of four components: (i) the location $s^{loc} = (x, y, z)$, (ii) the orientation represented by a rotation angle $s^{rot}$ within the XY plane, (iii) location description, and (iv) surrounding object descriptions. In our setup, we first sample the location and orientation considering four scenarios: (i) standable area on the floor with arbitrary viewpoint, (ii) sittable area with front viewpoint when sitting, (iii) reachable area of large objects (*e.g.*, cabinets and fridge) with viewpoints facing or against the object, and (iv) reachable area of small objects (*e.g.*, trashcan) with viewpoints directing standing point to object centers. We then generate location descriptions according to the interaction types (*e.g.*, "I'm standing on/sitting on/in front of the fridge ..."). For surrounding object descriptions, we first calculate the spatial relations between the location and surrounding objects, including distance, coarse direction (*e.g.*, *left*), and fine-grained relative direction (*e.g.*, *2 o'clock*). We then utilize these spatial relations to prompt GPT-3.5 for surrounding object descriptions. We provide more details and illustrative examples of sampled situations in Fig. 8.

**QA pairs generation**    Similar to prior works [28, 30], we adopt scene graphs to prompt LLM for data generation. We first instantiate each object in the scene graph with their attributes obtained by prompting GPT-4V [45] using the cropped object images. We then perform pair-wise calculations among the initialized objects to derive relations, which can be categorized into five types: in-contact vertical relations (*e.g.*, support), non-contact vertical relations (*e.g.*, above), horizontal distance (*e.g.*, near), horizontal proximity relations (*e.g.*, right) and multi-object relations (*e.g.*, between).

Table 1: **A comparison between MSQA and existing 3D vision-language datasets**. "Situated" indicates tasks with situation conditions. "Multi-modal" indicates whether the question contains multi-modal input. [†] indicates we only consider the proportion of newly collected data.

| Dataset | Situated | Multi-modal | Text collection | Quality Check | LLM scoring | Data sources | #Scenes | #Data |
|---|---|---|---|---|---|---|---|---|
| ScanQA [6] | ✗ | ✗ | human | ✓ | ✗ | ScanNet | 800 | 41k |
| 3D-QA [66] | ✗ | ✗ | human | ✓ | ✗ | ScanNet | 806 | 5.8k |
| Scan2Cap [14] | ✗ | ✗ | human | ✓ | ✗ | ScanNet | 800 | 41k |
| ScanScribe[†] [72] | ✗ | ✗ | template | ✗ | ✗ | 3RScan | 1185 | 90k |
| 3D-LLM[†] [24] | ✗ | ✗ | LLM-assisted | ✗ | ✗ | ScanNet, Habitat-Matterport [51] | 1.2k | – |
| LEO[†] [28] | ✗ | ✗ | LLM-assisted | ✗ | ✗ | 3RScan | 1185 | 191k |
| SQA3D [41] | ✓ | ✗ | human | ✓ | ✗ | ScanNet | 650 | 33.4k |
| MSQA | ✓ | ✓ | LLM-assisted | ✓ | ✓ | ScanNet, 3RScan, ARKitScenes | **1734** | **251K** |

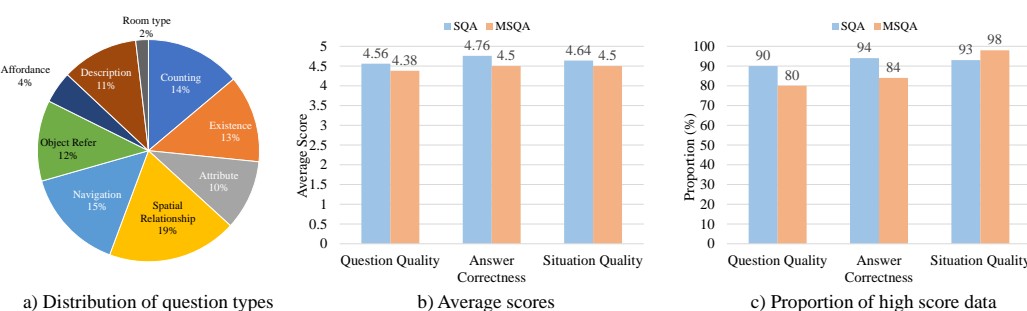

a) Distribution of question types  b) Average scores  c) Proportion of high score data

Figure 4: **Dataset statistics and quality evaluation**. We visualize (a) the distribution of question types in MSQA, (b) average quality scores of MSQA, and (c) the proportion of high-scoring data compared with SQA3D.

After establishing these relationships as edges in the scene graphs, we adjust the horizontal proximity relations according to the location and viewpoint of the sampled situation to obtain situated scene graphs. With these situated scene graphs, we design system prompts and hand-crafted examples to prompt GPT-3.5 [44] to generate situated question-answer pairs. We focus on 9 distinct question scopes, spanning object attributes, counting, spatial relationships, navigation actions, *etc*. (as shown in Fig. 4(a)). During prompting, we instruct the LLM to output question categories. To further enhance the diversity of LLM-generated QA pairs, we use various combinations of seed examples and sample various situated sub-scene-graphs conditioning on different considered distances for question generation. We provide more details for QA pair generation in Appendix A.2.

**Data refinement**  To enhance the quality of the generated situated question-answer pairs, we conduct a refinement procedure encompassing two main aspects: (1) for the situated scene graphs, we examine the distribution of attributes and relations to mitigate any potential bias that could lead to hallucination, and (2) we manually review the LLM-generated QA pairs to validate their accuracy and devise filtering functions based on regular expression to detect and correct potential errors. Illustrative examples of the refinement procedures are provided in Appendix A.3.1. As prior works [28, 68] have highlighted the importance of data balancing, we balance the answer distribution of generated data by filtering out imbalanced question-answer pairs. Through these procedures, we collect 251K multi-modal situated QA pairs across ScanNet, 3RScan, and ARKitScenes. We provide a comparison between MSQA and existing datasets in Tab. 1 and more statistics in Appendix A.6.

## 3.2 Data Quality Control

Despite the scalability of the LLM-based data collection pipeline, the quality of generated data has raised major concerns, especially in 3D vision-language tasks where grounding language is challenging. To address these concerns, we conduct a human study comparing our generated data to human-annotated data in SQA3D. Specifically, we sample 100 data instances from MSQA and SQA3D and mix them for human assessment. The human evaluators are instructed to score the data on three aspects: (1) the naturalness and clarity of situation descriptions, (2) the situational dependence and clarity of questions, and (3) the accuracy and completeness of answers. Each aspect was rated on a scale from 1 to 5. Detailed information about the evaluation workflow is provided in Appendix B. The evaluation results, shown in Fig. 4(b), indicate that MSQA's quality is comparable to SQA3D across all aspects. Additionally, Fig. 4(c) shows that the proportion of high-scoring data (*i.e.*, quality with score $\geqslant 4$) in MSQA matches or exceeds that of SQA3D. This highlights the quality of MSQA and also the effectiveness of our data refinement procedures.

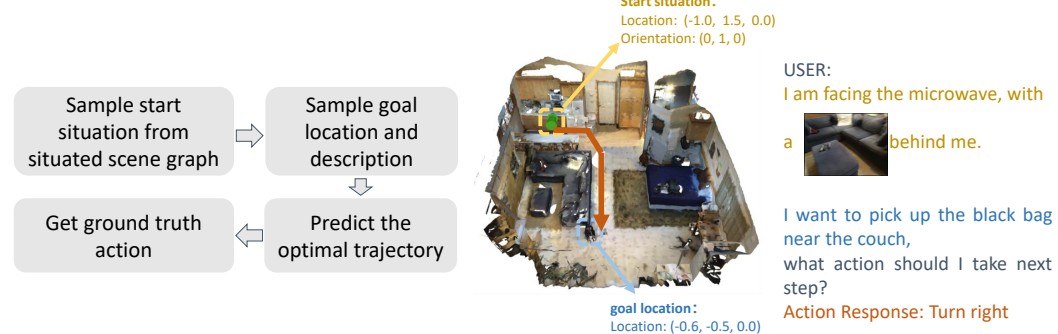

Figure 5: **The generation pipeline of the multi-modal situated next-step navigation (MSNN) task.** We follow a generation pipeline similar to QA pairs for situated navigation action.

## 4 Evaluation Benchmarks

In this section, we give a detailed description of the evaluation tasks considered for multi-modal situated reasoning. Specifically, we consider the following two benchmarking tasks:

**Multi-modal Situated Question Answering (MSQA)**    As mentioned Sec. 3, we evaluate models' capability in situation awareness and handling interleaved multi-modal input in MSQA. Specifically, given a multi-modal situation description, the model answers a text-image interleaved question grounded in the 3D scene. Since the responses are open-ended, former metrics, such as classification accuracy and exact-match accuracy can not give a correct evaluation. To solve this problem, we use a GPT-based evaluation metric for open-ended responses following OpenEQA [42] and extend its prompt sets for 3D situated reasoning (see detailed prompt in Appendix B.1.1). Above all, we report the correctness score $C$ for the test set with $N$ samples following OpenEQA, $C$ could be calculated by:

$$C = \frac{1}{N} \sum_{i=1}^{N} \frac{s_i - 1}{4} \times 100\%,$$

where $s_i$ (ranging from 1 to 5, the higher the better) is generated by the LLM when prompted with the question, ground truth answer, and the model response.

**Multi-modal Situated Next-step Navigation (MSNN)**    In addition to MSQA, we also aim to evaluate the models' capability of situation awareness through embodied AI tasks such as navigation. To separate long-horizon planning from situated understanding, we propose the MSNN task, which focuses on predicting the best immediate next step action grounded by the current situation and navigation target in a 3D scene. Specifically, given the agent's current interleaved multi-modal description of the situation (*i.e.*, location, orientation, and text description), textual goal description, and the overall scene, we instruct models to answer the immediate next action for navigating to the goal in a textual form. For evaluation, we generate MSNN data following a pipeline similar to situated QA pair generation with four critical components: (1) starting situation sampling, (2) goal sampling, (3) optimal trajectory prediction, and (4) calculation of ground truth immediate next-step action. The optimal trajectory is sampled by running an A* algorithm planning the shortest path from the starting location to the goal on the floor plan and the immediate next-step action is determined by following the direction of optimal trajectory relative to the starting situation. In total, we generate a dataset comprising 34K MSNN data samples across 378 3D scenes in ScanNet. This dataset is further utilized for supervised fine-tuning and MSNN evaluation. We provide more details on MSNN data generation and data statistics in the *Appendix*.

## 5 Experiment

### 5.1 Model Settings

Inspired by recent advancement in 3D generalist models, LLMs and VLMs, we propose several potential approaches for MSQA and MSNN including models that can be directly applied to these tasks in a zero-shot setting, and models that require instruction tuning.

**Zero-shot models** We investigate the ability of existing LLMs and VLMs (*i.e.*, GPT-3.5 [44] and GPT-4o [45]) for multi-modal situated reasoning. Recognizing the limitation of these models in handling 3D point clouds, we provide these models with textual descriptions of the 3D scenes as inputs. Specifically, the scene is described as a collection of objects, with each object characterized by its category, location, size, and attributes. This textual description of the scene is then integrated with the interleaved multi-modal situation descriptions, instructions, and questions, and further processed by the LLM or VLM. For text-only models (*i.e.*, LLMs), we substitute images of objects with their corresponding object categories as model input. We also incorporate Claude-3.5-Sonnet [5] to eliminate the potential bias within the GPT family.

**Instruction tuning** Following recent advancement in 3D generalist models [24, 28], we fine-tune existing 3D vision-language foundation models on MSQA and MSNN. In particular, we choose LEO [28] as a representative model given its superior performance in 3D VL understanding and reasoning. Since LEO does not naturally support interleaved multi-modal input, we modify the input by replacing the input images with their corresponding object categories similar to zero-shot models. Additionally, we extend LEO to accommodate the interleaved multi-modal input setting, resulting in our strong baseline model, MSR3D, tailored for situated reasoning and navigation. MSR3D deliberately models the situation by translating and rotating the point cloud input conditioned on the agent's situation. We choose MSR3D as our primary model for subsequent ablation studies and analyses. We provide more details on the design of MSR3D in Appendix C.

## 5.2 Evaluation Results

In this section, we provide evaluation results of models on MSQA and MSNN. We report the average correctness score (as illustrated in Sec. 4) across test sets for both tasks. Additionally, we consider different settings on the modality of the situation and question input (*Input*), the representation of 3D scenes (*Scene*), and the model setting (*Setting*). For MSNN, we ablate the choice of pre-training data (*PT data*) as an additional axis to verify the usefulness of MSQA for embodied tasks.

### 5.2.1 Multi-modal Situated Question Answering (MSQA)

We present the experimental results of MSQA in Tab. 2 and report the following findings:

**Zero-shot models struggle in situated spatial reasoning.** Zero-shot models excel in answering commonsense questions, such as those related to affordance and room type (categorized as *Other*), likely due to LLMs' proficiency in natural language tasks. Given that object attributes are provided in the list, these models show superior performance in attributes and descriptions compared to fine-tuned models. However, they fall short in addressing spatial relationships and navigation questions, highlighting their limitations in multi-modal situated reasoning.

**Situation modeling matters in situated spatial reasoning.** 3D vision-language models like LEO struggle without fine-tuning on MSQA, reflecting its limitations as a generalist foundation model. Our model trained without interleaved input outperforms LEO on spatial relationships and navigation, highlighting the importance of our situation modeling method. Meanwhile, the performance of MSR3D declines sharply in fine-tuning without 3D scene input (blind). This underscores the importance of situation awareness and 3D scene understanding in addressing MSQA.

**3D point cloud is a better scene representation compared to textual descriptions.** We conduct an additional experiment with solely textual descriptions, which are derived by prompting GPT-3.5 based on situated scene graphs. The situations used for generating textual descriptions are the same as those for QA pairs in MSQA. See examples of textual descriptions in Appendix A.3.3. The results in Tab. 2 (row "DES") indicate a notable drop when provided with textual descriptions, especially in object attribute, spatial relation, and navigation. To proceed, we probe the reason why "DES" shows better performance in counting. As shown in Tab. 3, "DES" is better for GT$< 3$ but worse for GT$\geqslant 3$. This is intuitive since "DES" explicitly depicts the target objects in the input. However, when the count of target objects exceeds a certain threshold, some target objects are likely to be truncated due to limited context length. In summary, the results demonstrate that the 3D point cloud serves as a more efficient representation for situated reasoning compared to textual descriptions.

**Situation component matters for situated reasoning.** To reveal the effectiveness of the situation for FT models, we add an FT baseline with the situation component entirely removed, retaining the 3D scene and question as input. The results in Tab. 2 (w/o situation) show a notable drop in performances

Table 2: **Experimental results on MSQA**. We use *Attr.* for question categories including object attributes and descriptions, *Spatial* for spatial relationship and object referral, and *Others* for affordance and room type. "FT" denotes models fine-tuned on MSQA, "T+I" for the interleaved text-image input, and "PCD/OBJ" for scene point cloud and object attribute list, respectively. *Input* includes both situation and question except the last row.

| Model | Input | Scene | Setting | Count. | Exist. | Attr. | Spatial | Navi. | Others | Overall |
|---|---|---|---|---|---|---|---|---|---|---|
| GPT-3.5 | T | OBJ | zero-shot | 34.84 | 74.48 | **75.77** | 27.86 | 42.95 | 88.02 | 50.65 |
| GPT-4o | T+I | OBJ | zero-shot | 31.20 | 71.41 | 75.21 | 31.50 | 36.67 | **88.03** | 49.68 |
| Claude-3.5-Sonnet | T+I | OBJ | zero-shot | 32.57 | 66.28 | 69.88 | 30.10 | 45.48 | 83.61 | 49.73 |
| LEO | T | PCD | zero-shot | 0.79 | 15.51 | 11.83 | 7.27 | 2.31 | 15.34 | 7.84 |
| | T | PCD | FT | **36.22** | **88.46** | 51.66 | 46.88 | 56.82 | 80.25 | 55.86 |
| MSR3D | T+I | PCD | FT (blind) | 11.91 | 31.02 | 20.57 | 22.26 | 25.77 | 34.45 | 22.92 |
| | T | PCD | FT | 33.46 | 87.45 | 53.65 | **48.91** | 61.89 | 75.00 | **56.48** |
| | T+I | PCD | FT | 33.46 | 86.28 | 50.88 | 42.79 | **62.56** | 73.31 | 54.13 |
| | T+I | DES | FT | 35.82 | 88.20 | 43.91 | 35.52 | 52.42 | 73.10 | 50.05 |
| | T+I | PCD | FT (w/o situation) | 30.78 | 85.51 | 45.35 | 42.66 | 52.97 | 71.00 | 51.20 |

Table 3: Comparison of counting performance between point cloud input (PCD) and textual description input (DES).

| Input | Overall | GT=1 | GT=2 | GT=3 | GT=4 | GT=5 |
|---|---|---|---|---|---|---|
| PCD | 33.46 | 21.62 | 61.36 | 36.73 | 3.22 | 14.29 |
| DES | 35.82 | 32.43 | 82.95 | 10.20 | 0 | 0 |
| Δ | 2.36 | 10.81 | 21.59 | -26.53 | -3.22 | -14.29 |

Table 4: Interleaved multi-modal input (T+I) *vs.* text-only input (T), on subsets where images appear in either situation (**S**) or question (**q**).

| Input | **S** w/ img, **q** w/o img | **S** w/o img, **q** w/ img |
|---|---|---|
| T | 55.54 | 56.41 |
| T+I | 56.48 | 43.58 |
| Δ | 0.94 | -12.83 |

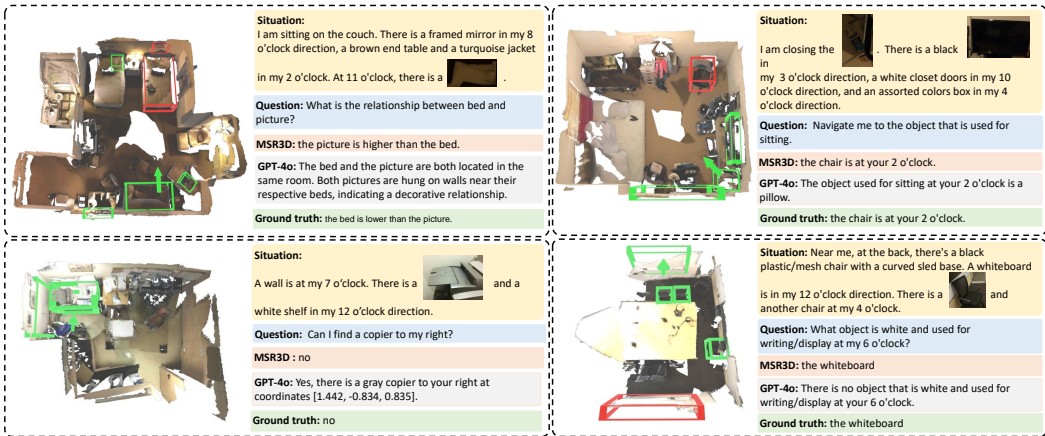

Figure 6: **Qualitative visualization on MSQA**. Top left: spatial relationship. Top right: navigation. Bottom left: object existence. Bottom right: object refer.

after removing the situation component. In particular, the drop in questions related to navigation is more salient, which echoes the evaluation results in MSNN and highlights the importance of the situation component. More analyses can be found in Appendix D.3.

**Interleaved multi-modal input introduces new challenges for situated reasoning.** Despite the advantages of interleaved multi-modal input, we observe that MSR3D (T+I) shows a slightly inferior performance compared to text-only input (T). To investigate this subtle difference, we extract two subsets from the test set by making the images only appear in either situation or question. The evaluation results on these two subsets are reported in Tab. 4, which indicate that "T+I" suffers a significant drop in the subset where the images only appear in question. We conjecture that incorporating images in question strengthens the challenge of situated reasoning probably because identifying the queried objects from images requires extra grounding ability.

### 5.2.2 Multi-modal Situated Next-step Navigation (MSNN)

We present the experimental results of MSNN in Tab. 5 and report the following findings:

Table 5: **Experimental results on MSNN**. We use the same notations for input, scene, and settings following Tab. 2. LEO-align is the pre-training dataset proposed in [28].

| Model | Input | Scene | Settings | PT data | Accuracy |
|---|---|---|---|---|---|
| GPT-3.5 | T | OBJ | zero-shot | – | 20.1 |
| GPT-4o | T+I | OBJ | zero-shot | – | 27.55 |
| LEO | T | PCD | FT | LEO-align | 31.44 |
|  | T | PCD | FT | MSQA | 37.05 |
| MSR3D | T+I | PCD | FT | – | 45.46 |
|  | T | PCD | FT | MSQA | 45.61 |
|  | T+I | PCD | FT (w/o situation) | MSQA | 31.66 |
|  | T+I | PCD | FT | MSQA | **48.4** |

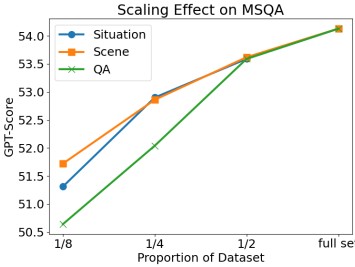

Figure 7: **Scaling effect on MSQA** with different pretraining data scales.

**MSNN is challenging.** The results in Tab. 5 indicate that both state-of-the-art LLMs (*i.e.*, GPT-3.5 and GPT-4o) and 3D VL models encounter considerable challenges in solving MSNN. This implies the value of the proposed MSNN task for 3D situated reasoning and embodied AI.

**MSQA is beneficial as a pretraining source for embodied AI.** We find that adopting MSQA for pretraining (both LEO and MSR3D) significantly improves the performances on MSNN, which indicates the effectiveness of MSQA as a pretraining source for addressing embodied navigation.

**Situation modeling of MSR3D is effective.** We find that MSR3D (T), endowed with situation modeling, shows a significantly higher accuracy in navigation action prediction (+8.56%) compared with LEO (T). This demonstrates the effectiveness of our situation modeling method. Additionally, we test MSR3D without situation by masking the location and orientation of the agent, which leads to a great performance drop as shown in Tab. 5 (w/o situation). Such a drop demonstrates the critical role of situation information and that MSR3D can utilize the situation information well.

## 5.3 Additional Analysis

**Scaling effect**  We explore the scaling effect on MSQA by training MSR3D with different data scales. We investigate three factors for scaling: QA (randomly downsampling QA pairs), situation (downsampling both QA pairs and situations), and scene (downsampling both QA pairs and scenes). As shown in Fig. 7, we observe a consistent trend of improvement when scaling up along the three factors, which exhibits a significant scaling effect and manifests the potential of further scaling up. We also provide additional analysis of the scaling effect on the MSNN task in Appendix D.1.

**Cross-domain transfer**  We divide the MSQA data into three subsets according to the scene domain: ScanNet [17], 3RScan [60] and ARK-itScenes [7]. Then we investigate cross-domain transfer by training MSR3D on each subset and evaluating on all the subsets, respectively. The results in Tab. 6 show that the best performance on each subset is achieved by in-domain training

Table 6: **Experiments on cross-domain transfer**. The left column denotes the source domain for training, and the top row denotes the target domain to transfer.

|  | ScanNet | 3RScan | ARKitScenes |
|---|---|---|---|
| ScanNet | **55.40** | 43.33 | 52.16 |
| 3RScan | 50.15 | **45.08** | 55.35 |
| ARKitScenes | 40.08 | 40.07 | **63.34** |

(**bold**) rather than cross-domain transfer, showcasing the domain gap. And training on ARKitScenes elicits inferior cross-domain transfer results. Considering the relatively simple scenes in ARKitScenes, it implies that training on complex scenes would be beneficial for cross-domain generalization.

## 6 Conclusion

In this paper, we introduce Multi-modal Situated Question Answering (MSQA), a large-scale multi-modal situated reasoning dataset collected with a scalable data generation pipeline. MSQA comprises 251K situated QA pairs across a variety of real-world scenes, presented in a unified format with interleaved text, images, and point clouds. We present a challenging benchmark based on MSQA for evaluating multi-modal situated reasoning in 3D scenes. Additionally, we propose Multi-modal Situated Next-step Navigation (MSNN), a task to assess the capability of situated reasoning and embodied navigation. Our comprehensive experiments highlight the value of our dataset and benchmarks. We hope this work will advance the development of situated scene understanding and embodied AI.

**Limitations and future work.** Our work proposes an automatic pipeline to scale up multi-modal situated reasoning data based on existing real-world 3D assets. We also introduce an innovative evaluation task MSNN for situated reasoning and embodied navigation. Despite our contributions, some limitations remain to be addressed.

Firstly, LLM-generated data needs further alignment with human preference to achieve higher data quality. Despite our meticulous design in refinement procedures and data balance, some unnatural data remains due to the rule-based scene graph and biases of LLMs. For instance, LLMs may select distant objects for situational descriptions, which might be an improbable behavior for humans. We encourage further exploration of human feedback integration in the data generation process to better align with human preference.

Secondly, we have not yet fully leveraged the available 3D assets. Expanding our data generation pipeline to cover more real-world and synthetic 3D scenes will further enhance the scale and diversity of the situated reasoning data, probably inducing stronger models. Given the expense of creating large-scale QA pairs by prompting LLMs with situated scene graphs, we anticipate that training a specific LLM tailored for generating QA pairs from situated scene graphs could substantially reduce the cost of data generation. We leave this path for future research.

Finally, the evaluation tasks for assessing situational awareness and situated reasoning should not be confined to question answering and action prediction. For example, some other tasks focusing on scene understanding like object grounding could also be considered. We would explore more evaluation suites in future work.

## Acknowledgements

The authors thank Qing Li and Yixin Chen for providing valuable feedback, and Haoyu Shao for participating in data quality control.

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

# A Dataset Details

## A.1 Situation Sampling

The sampling rules of location and orientation are as follows.

- **Standing.** We evenly sample a point from the floor area as the location and an angle within $[0, 2\pi)$ as the orientation.
- **Sitting.** We randomly sample a point from the sitable area, *e.g.*, chairs and couches. The orientation is calculated based on the normal direction of the backrest.
- **Interacting with large objects.** For large objects, *e.g.*, cabinets and refrigerators, we first parse the interactive part such as the door. Then we sample a standing point from the nearby floor as the location and use the normal direction of the interactive part as the orientation.
- **Interacting with small objects.** For small objects, *e.g.*, bags and trash cans, we first sample a standing point from the nearby floor as the location and then use the direction from the standing point to the object center as the orientation.

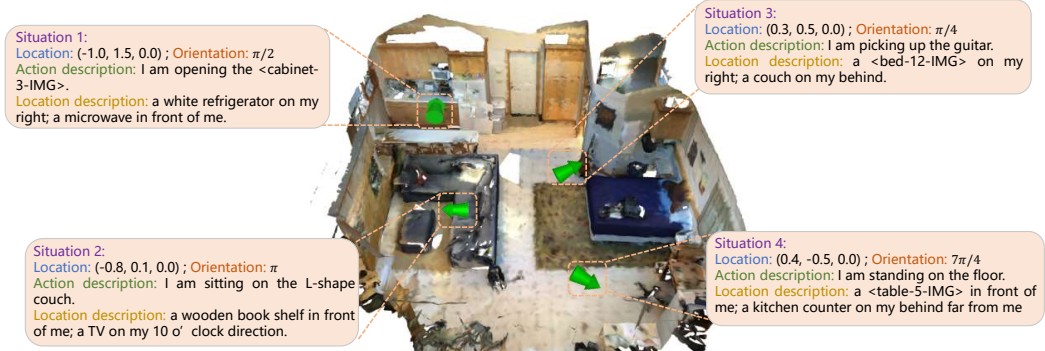

Figure 8: Examples of the situations. Each situation contains the location, orientation, action description and location description of the presumed agent. The image placeholders in the action and location descriptions, *e.g.*, "<beg-12-IMG>", will be replaced by a corresponding image.

## A.2 Data Collection

### A.2.1 HOI Examples

Table 7: Examples of HOI descriptions.

| Object | Generated Descriptions |
|--------|------------------------|
| door | I am opening the <door-<ID>-IMG>.
I am locking the <door-<ID>-IMG>.
I am closing the <door-<ID>-IMG>. |
| window | I am repairing the <window-<ID>-IMG>.
I am looking through the <window-<ID>-IMG>.
I am opening the <window-<ID>-IMG>. |
| picture | I am hanging the <picture-<ID>-IMG>.
I am photographing the <picture-<ID>-IMG>.
I am adjusting the <picture-<ID>-IMG>. |
| backpack | I am picking up the <backpack-<ID>-IMG>.
I am putting items in the <backpack-<ID>-IMG>.
I am wearing the <backpack-<ID>-IMG>. |

To boost diversity in situation descriptions, we create varied Human-Object Interaction (HOI) descriptions using LLMs. Tab. 7 displays HOI examples from ChatGPT [44] detailing interactions with both large and small objects. Object IDs correspond to specific instances. Overall, we have generated 4210 descriptions for 421 objects.

Table 8: Examples of the attributes detected by GPT-4V.

| Image | Concise attributes | Descriptive attributes |
|---|---|---|
|  | "color": "White and various colors from the view outside", "3D shape": "Rectangular prism", "material": "Glass and PVC or wood", "usage": "Allowing light in and providing view to the outside", "texture": "Smooth glass", "structure": "Paneled with a latch", "state": "Fixed installation" | "color": "The window frame is white, while the glass reflects the multicolored view of the outside world, including greens, blues, pinks, and other colors from the scenery and sky.", "3D shape": "The window is in the shape of a rectangular prism, typical for residential window construction.", "material": "The window frame appears to be made of either PVC or painted wood, while the window pane itself is likely made of clear glass.", "usage": "The window is used to let natural light into the room and provide the occupants with a visual connection to the outdoors, as well as potential ventilation.", "texture": "The glass of the window is smooth to the touch, providing a clear view outside.", "structure": "The window consists of a main, larger fixed pane with a patterned texture for privacy, and a smaller, possibly openable pane above it, equipped with a metallic latch possibly for opening and securing the window.", "state": "The window is a permanent part of the building's structure, installed in the wall, and appears to be closed at the moment." |
|  | "color": "white and grey", "3D shape": "rectangular prism", "material": "plastic and metal", "usage": "weighing", "texture": "smooth", "structure": "flat platform with digital display", "state": "stationary" | "color": "The object features a predominantly white color with grey accents.", "3D shape": "The object has a three-dimensional rectangular prism shape with a flat top surface.", "material": "The object appears to be constructed from a combination of plastic for the outer casing and possibly metal for the weighing mechanism.", "usage": "The object is used for measuring weight, typically that of a person standing on it.", "texture": "The surface texture of the object looks smooth to facilitate easy standing and cleaning.", "structure": "The object has a structured flat platform designed for standing upon, with a digital display panel for reading measurements.", "state": "The object is in a stationary state, placed on the floor, likely not in use at the moment." |
|  | "color": "light brown", "3D shape": "rectangular prism", "material": "wood", "usage": "holding items", "texture": "smooth", "structure": "four legs, a flat top, storage shelf", "state": "in use" | "color": "A light brown with possible variations due to lighting and shadows", "3D shape": "A primarily rectangular prism shape consistent with typical tables", "material": "Appears to be made from a wood material or wood-like composite", "usage": "Used for placing, organizing, and storing various items such as papers, books, electronic devices, and personal objects", "texture": "The surface looks smooth with a likely varnished or laminated finish", "structure": "The table has a flat horizontal top supported by four vertical legs and includes an under-table storage shelf for additional items", "state": "The table is currently in use, with various objects placed on top and the storage shelf being partially occupied" |

### A.2.2 Object Attribute Detection

We utilize GPT-4V for object attribute detection. Seven attribute types, *i.e.*, color, 3D shape, material, usage, texture, structure, and state, are considered. We first crop the object from the corresponding multi-view color images provided in each dataset, and then use the largest-size cropped image for prompting. The multi-view images are deblurred to enhance the image quality before cropping. Since GPT-4V is powerful enough to generate detailed attribute descriptions, besides concise attributes, we also prompt GPT-4V to generate the descriptive attributes of the object at the same time. The detailed prompt for attribute detection is illustrated in Fig. 9. Some examples of the concise and descriptive attributes are shown in Tab. 8.

### A.3 LLM-assisted Data Generation

Fig. 10 illustrates the prompts used to generate data. In particular, we instruct ChatGPT [44] to produce both the question-answer pairs and their corresponding types. We also provide a demonstration in Fig. 11.

### A.3.1 Refinement

Table 9: Quality of data produced by LLM- assisted generation methods, measured by answer accuracy on three categories of generated questions.

| Question Type | Counting | Existence | Non-existence |
|---|---|---|---|
| Accuracy(%) | 80.95 | 83.38 | 69.94 |

```
output_format = '{"short" : "color" :, "3D shape" :, "material":, "usage":, "texture", "structure":,
"state":,}, "long":{"color" :, "3D shape" :, "material":, "usage":, "texture", "structure":, "state":,}}'
object_name = sample["object_name"]
object_image = encode_image(sample["image_path"])

prompt = f"""
describe the color, 3D shape, material, usage, texture, structure, state of the object in the image.
the object maybe a {object_name}. please directly give a long and short description of the object
seperately in python dict format. the output should be like {output_format}, and can be parsed by eval
function.
"""

messages = [{"role": "user", "content": [{"type": "text", "text": prompt},
{"type": "image_url", "image_url": {"url": f"data:image/jpeg;base64,{object_image}"}}]}]
```

Figure 9: Prompt messages for GPT-4V for attribute detection.

```
messages = [{'role': 'system', 'content': 'Task:
Generate QA pairs based on a detailed scene graph represented in a dictionary. This dictionary includes
objects in four directions: front, left, back, and right, each further categorized by distance (far,
middle, near). The task involves creating questions and answers that reflect the spatial and relational
aspects of these objects from a first-person perspective.
Dictionary Structure:
{ "front": { "far": { /* Objects and attributes in the far distance in the 'front' direction */ },
"middle": { /* Objects and attributes in the middle distance in the 'front' direction */ },"near": { /*
Objects and attributes in the near distance in the 'front' direction */ }},"left": { /* Similar structure
for objects to the 'left' */ },"back": { /* Similar structure for objects in the 'back' */ },
    "right": { /* Similar structure for objects to the 'right' */ }
}
(Adapt to the specific content of each dictionary)
QA Pair Generation Rules:
1. QA Types:
- Attribute (color/shape/material/texture): Queries about specific attributes of objects.
- Counting: Inquiries regarding the number of objects or specific features.
- Existence: Determines whether an object is present or not.
- Affordance: Questions about the intended use or function of an object.
- Spatial Relationship: Explores the spatial arrangement between two or more objects.
- Room Type: Questions identifying or inferring the type of room based on the objects present.
- Object Refer (multiple/single): Identifies one or multiple objects.
- Description-Single: Asks for a detailed description of a single object.
- Attribute-<attribute>-Multiple: Inquires about specific attributes of multiple objects.
- Direction-Distance: Comparing the relative distances of objects.
- Direction-Based Navigation: Navigating towards or finding the direction of a specific object.
2. First-Person Perspective with Distance: Ensure all questions include a first-person directional
description (e.g., "in front of me," "to my left") and consider the distance (far, middle, near) of
objects.
3. Format:
- Questions (Q) should reference objects' locations and attributes from a first-person perspective using
the "<<label>-<id>-M>" format.
- Queried Objects (T) should be "<label>-<id>".
- Answers (A) should be concise, directly based on the dictionary. If information is not available,
respond with "unknown". Do not include object IDs in answers.
- Specify the Type of question (Type) from the listed QA types.
Goal: Generate 15 accurate and relevant QA pairs that utilize the dictionary to address the spatial
layout and characteristics of objects. The questions should consider the direction, distance, and
specific object formatting while adhering to the first-person perspective.
'}]

for sample in few_shot_samples:
    messages.append({'role': 'user', 'content': sample['content']})
    messages.append({'role': 'assistant', 'content': sample['response']})
messages.append({'role': 'user', 'content': sample['query']})
```

Figure 10: Prompts for ChatGPT.

Despite ChatGPT's advanced reasoning capability, it sometimes generates inaccurate data. To improve quality assurance, the refinement procedure serves as an extra safeguard. We manually corrected errors in three types: counting, existence, and non-existence, and checked other types for quality and reliability. We also employed filtering methods to remove unreasonable data points, like

```
sample['content'] = ''{'front': {'far': {}, 'middle': {'lamp-21': {'attributes': {'color': 'pink', '3D
shape': 'curved', 'usage': 'lighting', 'texture': 'smooth', 'structure': 'l-shaped with a base and an
arm', 'state': 'on'}}, 'tissue box-22': {}}, 'near': {'ball-10': {'attributes': {'color': 'turquoise',
'3D shape': 'sphere', 'material': 'rubber', 'usage': 'toy', 'texture': 'bumpy', 'structure': 'solid',
'state': 'intact'}}}}, 'left': {'far': {}, 'middle': {}, 'near': {'tissue box-9': {'attributes':
{'color': 'white', '3D shape': 'rectangular prism', 'material': 'cardboard', 'usage': 'dispensing
tissues', 'texture': 'smooth', 'structure': 'box with an opening at the top', 'state': 'intact'}}, 'wall-
1': {}, 'bag-8': {}}}, 'back': {'far': {}, 'middle': {'door-12': {'attributes': {'color': 'white', '3D
shape': 'rectangular prism', 'material': 'wood', 'usage': 'entrance or privacy', 'texture': 'smooth',
'structure': 'solid', 'state': 'closed'}}, 'closet doors-7': {'attributes': {'color': 'off-white', '3D
shape': 'rectangular prisms', 'material': 'wood or wood product', 'usage': \"to conceal or protect a
closet's contents\", 'texture': 'smooth', 'structure': 'flat panels with handles', 'state': 'closed or
partially closed'}}, 'doorframe-13': {}}, 'near': {}}, 'right': {'far': {}, 'middle': {'ottoman-19':
{'relations': ['support ball-20'], 'attributes': {'color': 'brown', '3D shape': 'rectangular prism',
'material': 'fabric, wood', 'usage': 'storage, seating', 'texture': 'soft, smooth', 'structure': 'box
with lid, legs', 'state': 'open, in use'}}, 'wall-14': {}, 'ball-20': {'relations': ['supported by
ottoman-19']}, 'pillow-18': {'attributes': {'color': 'white with stripes', '3D shape': 'cylindrical',
'material': 'fabric', 'usage': 'comfort or decoration', 'texture': 'soft', 'structure': 'tubular with
ends closed', 'state': 'intact'}}, 'picture-15': {}, 'curtain-2': {}, 'wall-16': {}, 'window-17': {}},
'near': {'bed-5': {'attributes': {'color': 'white', '3D shape': 'rectangular prism', 'material': 'wood,
fabric, metal', 'usage': 'sleeping or resting', 'texture': 'soft and smooth', 'structure': 'frame,
mattress, bedding', 'state': 'made'}}}}}''

sample['response'] = ''Q: Is there a <pillow-18-M> on my right that is white with stripes?\nT: pillow-
18\nA: yes\nType: existence\n\nQ: What is supporting the <ball-20-M> on my right?\nT: ball-20\nA:
ottoman\nType: spatial relationship\n\nQ: How can I reach the <door-12-M> behind me?\nT: door-12\nA: Turn
around and walk to the middle distance.\nType: direction-based navigation\n\nQ: What is the color of the
<bed-5-M> near me on my right?\nT: bed-5\nA: white\nType: attribute-color\n\nQ: How many items are in the
middle distance in front of me?\nT: lamp-21, tissue box-22\nA: two\nType: counting\n\nQ: What is the
usage of the <ottoman-19-M> on my right?\nT: ottoman-19\nA: storage and seating\nType: affordance\n\nQ:
Can I find a <ball-10-M> near me in front?\nT: ball-10\nA: no\nType: existence\n\nQ: What texture is the
<tissue box-9-M> to my left?\nT: tissue box-9\nA: smooth\nType: attribute-texture\n\nQ: From where I am,
how do I get to the <tissue box-9-M> on my left?\nT: tissue box-9\nA: Turn to your left and walk a short
distance.\nType: direction-based navigation\n\nQ: What object is pink and in front of me?\nT: lamp-21\nA:
The lamp.\nType: object refer\n\nQ: What is on the <ottoman-19-M> on my right?\nT: ottoman-19\nA: a
ball\nType: object refer\n\nQ: How many objects are designed for sitting or resting in this room?\nT:
ottoman-19, bed-5\nA: two\nType: counting\n\nQ: Is there a <window-17-M> on my right?\nT: window-17\nA:
yes\nType: existence\n\nQ: What is the structure of the <lamp-21-M> in front of me?\nT: lamp-21\nA: l-
shaped with a base and an arm\nType: attribute-structure\n\nQ: What direction should I look to find
something for lighting?\nT: lamp-21\nA: In front of you at a middle distance.\nType: direction-based
navigation''
```

Figure 11: An example of human demonstration.

Table 10: Examples of refinement.

| Types | Raw Responses | Refined Responses |
|---|---|---|
| Counting | Q: How many chairs are on my right?
A: 5 | Q: How many chairs are on my right?
A: 4 |
| Existence | Q: Is there a table on my 6 o'clock?
A: yes | Q: Is there a table on my 6 o'clock?
A: no |
| Non-existence | Q: Is there a book in the room?
A: no | Q: Is there a book in the room?
A: yes |
| Negative responses | Q: What is the state of the tv?
A: unknown | *The negative response will be removed.* |

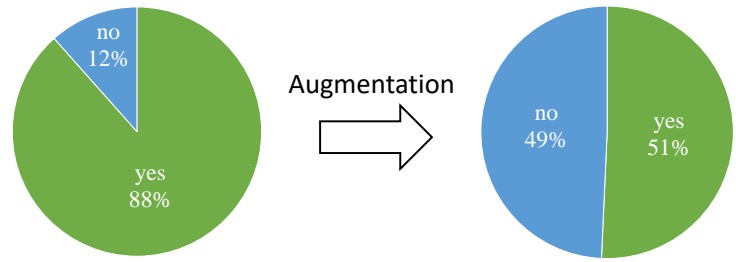

Figure 12: Distribution of answer for existence before and after data augmentation.

those with answers stating "not specified attributes are in the scene graph"/"unknown", etc. Tab. 9 shows the error statistics. Incorrect responses are adjusted according to the situated scene graphs, with examples in Tab. 10.

### A.3.2 Data Balance

Fig. 12 depicts the answer distribution for existence across three datasets. We enriched the QA pairs on existence by incorporating *no* responses, primarily from objects-in-the-scene or objects-in-the-wild. Using ChatGPT, we created 600 in-the-wild objects. Post-data balancing, the *yes/no* ratio nears 1:1 Fig. 12.

### A.3.3 Situated Scene Description

Table 11: Examples of situated scene description.

| 3D Scene | Situated Scene Description |
|---|---|
| 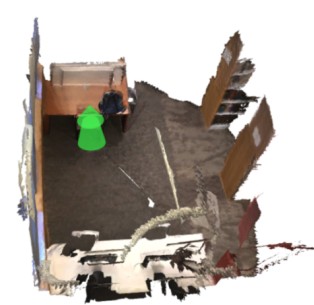 | Adjacent to a brown wooden divider resting on the table, a gray fabric chair with wooden legs supports a black and silver monitor on its four legs. This setup combines comfort and technology, creating a cozy corner for work or leisure activities. Positioned on the smooth floor, a black rectangular prism-shaped plastic trash can stands to the right of a gray and black chair. The chair features textured and smooth fabric and wood materials, providing comfort for seating. This setup, with smooth textures and clean lines, offers a practical and sleek workstation for productivity. |
| 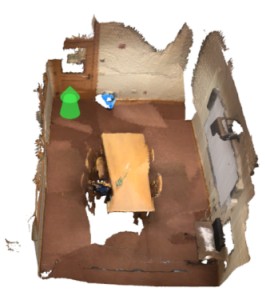 | A symphony of wooden chairs fills the space, with a solid brown chair to the left, a chair embedded into a light brown table, a wooden brown chair to the right, and a beige and brown chair close by. These pieces, each with their unique design and comfortable seating, come together to form a harmonious seating area, ideal for discussions or relaxation in a cozy and inviting setting. A solid brown wooden chair with a sleek backrest stands elegantly next to a brown wooden table, both pieces with a smooth texture and sturdy construction. Inside the chair rests a blue and black synthetic fabric backpack, adding a touch of color to the room. The blend of textures and colors in the chairs and table creates a warm and inviting atmosphere, perfect for gatherings or collaborative work sessions. |

The examples of situated scene description can be found in Tab. 11.

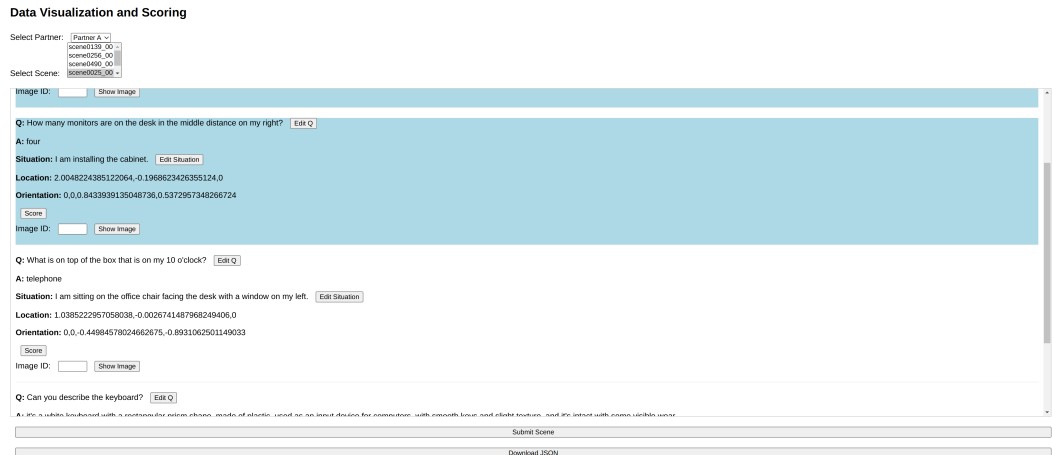

Figure 13: Data visualizer for human study.

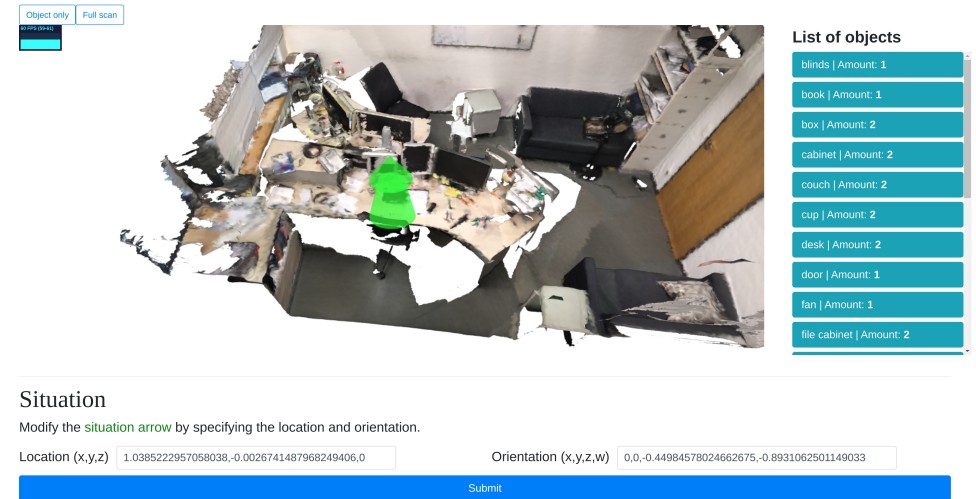

Figure 14: 3D viewer for human study.

## A.4 Human Study

We developed two web pages for conducting a human study. Using the data visualizer, partners can select corresponding scenes and data instances, and then provide scores for question quality, answer correctness, and situation quality. The orientation and location for each data instance appear in the data instance block. When the partner inputs these values, the 3D viewer automatically adjusts the location and orientation (as indicated by the arrow). The web pages are shown in Fig. 13 and Fig. 14. After scoring, we collect all scores and analyze the results for SQA3D and MSQA. We assess the data quality considering the following principles:

**Question.** A high-quality question should be situational, spatial-aware and unambiguous. For example, questions like: *How many brown wooden tables are near me on my right?/Is the door open on my left front me?* are both perfect examples.

**Situation.** Ideally, the situation description can locate the agent in the 3D scene uniquely. For example, a situation description: *There is a gray trash can on my right in a middle distance. I am sitting on a black chair.* In this description, the spatial relationship, distance and activity are clearly presented.

**Answer.** The correctness is crucial to guide the model to reason in 3D scenes. The questions with accurate and unique answers such as existence and counting can be scored according to the 3D scene,

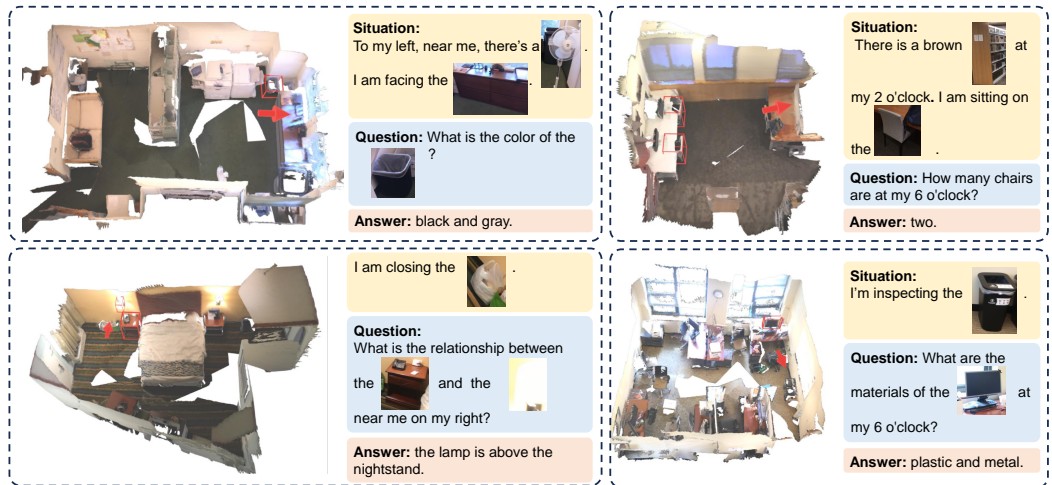

Figure 15: Examples of MSQA.

situation and question. For questions such as describing attributes for a queried object, correctness of description and richness of detail are both factors considered.

## A.5 Data Examples

We provide more examples of MSQA in Fig. 15.

## A.6 Dataset Statistics

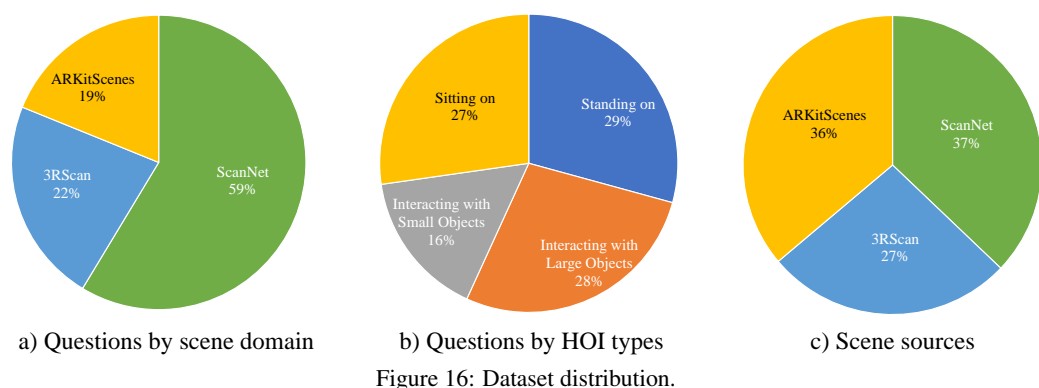

a) Questions by scene domain     b) Questions by HOI types     c) Scene sources

Figure 16: Dataset distribution.

In Fig. 16, we illustrate more statistical results of MSQA, including the distribution of questions considering HOI types, data source, and the distribution of scene quantity. Fine-grained statistics are shown in Tab. 12

## B Evaluation Details

### B.1 GPT-score

#### B.1.1 Prompts for GPT-score

We craft a precise prompt to evaluate open-ended replies in MSQA, depicted in Fig. 17. Specifically, we embed scoring examples to bolster robustness and synchronize the scores with human evaluation.

Table 12: Fine Grained Dataset Statistics of MSQA. $s$ denotes situation, $q$ denotes question, $a$ denotes answer.

| | ScanNet | | | 3RScan | | | ARKitScenes | | |
|---|---|---|---|---|---|---|---|---|---|
| | Train | Val | Test | Train | Val | Test | Train | Val | Test |
| Total $s^{txt}$ | 146011 | 891 | 832 | 55685 | 622 | 315 | 46632 | 634 | 266 |
| Total $s^{(loc,rot)}$ | 10988 | 668 | 639 | 3203 | 330 | 227 | 3491 | 403 | 183 |
| Total $s^{img}$ | 28001 | 137 | 154 | 13750 | 165 | 79 | 4473 | 57 | 21 |
| Total $q$ | 146011 | 891 | 832 | 55685 | 622 | 315 | 46632 | 634 | 266 |
| Total $q^{img}$ | 76691 | 452 | 566 | 24500 | 287 | 132 | 18444 | 240 | 103 |
| Scenes | 516 | 64 | 67 | 375 | 46 | 45 | 471 | 111 | 48 |
| Avg. Len. of $s^{txt}$ | 20.04 | 19.91 | 29.24 | 14.76 | 15.33 | 13.55 | 14.32 | 14.18 | 14.38 |
| Avg. Len. of $q$ | 10.28 | 10.33 | 10.16 | 9.57 | 9.62 | 9.54 | 10.04 | 10.17 | 10.11 |
| Avg. Num. of $s^{img}$ | 0.19 | 0.15 | 0.19 | 0.25 | 0.27 | 0.25 | 0.10 | 0.09 | 0.08 |
| Avg. Num. of $q^{img}$ | 0.53 | 0.51 | 0.68 | 0.44 | 0.46 | 0.42 | 0.40 | 0.38 | 0.39 |
| Avg. Len. of $a$ | 6.91 | 6.86 | 4.93 | 7.53 | 7.26 | 5.05 | 4.81 | 4.69 | 3.24 |

Score open-ended answers from 1 to 5 based on accuracy, completeness, and relevance to the ground truth.
Criteria:
Counting: Question: How many tables are in the room? Ground Truth: Three Example Response: Two Score: 1 (Significant discrepancy) Existence: Question: Is there a chair on my left? Ground Truth: Yes Example Response: Yes, there is a chair on the left. Score: 5 (Correct match) Description: Question: Describe the <couch-3-IMG> on my left. Ground Truth: Black couch with yellow and orange pillows. Example Response: Multicolored bed. Score: 1 (Incorrect identification) Spatial Relationship: Question: What is the relationship between the desk and the computer tower? Ground Truth: Inside the desk. Example Response: To the right of the desk. Score: 1 (Incorrect relationship) Question: What is the relationship between the chair and the table? Ground Truth: On the left of the table. Example Response: On the left of the table. Score: 5 (Correct match) Navigation: Question: How do I reach the window on my right from my current position? Ground Truth: Turn right and walk to the middle distance. Example Response: Turn to your right. Score: 3 (Partial instructions) Question: Where is the wooden chair located in relation to me? Ground Truth: At your 6 o'clock. Example Response: At your 7 o'clock. Score: 4 (Minor discrepancy) Example Response: At your 10 o'clock. Score: 1 (Major discrepancy) Object Reference: Question: What is on the table behind me? Ground Truth: A book, a pencil, and a monitor. Example Response: a pencil and a monitor. Score: 3 (Partial context) Room Type or Affordance: (Provide context-specific details and examples) Guidelines: Score 5: Perfect or highly accurate response. Score 1: Significant inaccuracies or discrepancies. Score 2-4: Reflect partial correctness or minor errors.
Output only the score.

Figure 17: Prompt for LLM-assisted scoring.

Table 13: Pearson coefficients between human scores and different metrics.

| | GPT-score(ours) | EM | EM-refined | BLUE-4 | METEOR | ROUGE |
|---|---|---|---|---|---|---|
| Correlation Coefficient | **0.9410** | 0.7083 | 0.7159 | 0.1913 | 0.7578 | 0.6633 |

Table 14: Examples of GPT-score, EM and EM-refined.

| Ground truth | Response | GPT-score(ours) | EM | EM-refined |
|---|---|---|---|---|
| it's a white sink with a rectangular box shape, made of ceramic, used for washing, with a smooth texture, intact, and it has a basin with a faucet and drain. | it's a white sink with a rectangular basin, made of ceramic, with a smooth texture, designed for washing, and it's clean. | 4 | 0 | 0 |
| a workspace or office area | an office or study room | 5 | 0 | 0 |
| two tables | two | 5 | 0 | 1 |

### B.1.2 Correlation of GPT-scores and Human Scores

We selected GPT-score as the metric for MSQA benchmark and crafted a detailed prompt to match human preferences. To confirm the soundness of this decision, we assessed responses and computed the **Pearson Correlation Coefficients** between human evaluations and various metrics. We manually scored 200 responses (randomly selected from the model's predictions) and calculated the corresponding metrics. Results are presented in Tab. 13. This table includes traditional VQA metrics like Exact Match, Refined Exact Match [28], and caption metrics such as BLEU-4, METEOR, and ROUGE. Findings show our metric closely correlates with human scores, whereas Exact Match and caption metrics show weaker correlations. We provide some qualitative examples of GPT-scoring in Tab. 14.

### B.2 MSNN Data Generation Details

The MSNN data generation pipeline can be separated into four parts, i.e., start situation sampling, goal sampling, optimal trajectory prediction and ground truth action calculation. Details about each part are stated as follows:

- **Start situation sampling.** We sample the start situations with the same sampling strategy stated in Section 3 of the paper. The location, orientation, and text-image interleaved descriptions are provided as the start situation.
- **Goal sampling.** We define the goal as navigating and interacting with one object in the scene. We first random sample an object in the scene and then generate a text description of the interaction description by prompting GPT-3.5 (see Fig. 18 for detailed prompts).
- **Optimal trajectory prediction.** With the sampled start and goal location, we can then predict the optimal navigation trajectory. Floor areas are regarded as the passable area for navigation, and the A* algorithm is employed to get the best navigation trajectory.
- **Ground truth action calculation.** After determining the optimal navigation trajectory from the starting point to the target destination, we subsequently determine the agent's immediate action by calculating the required orientation adjustment relative to the initial situation. We consider four potential actions, *i.e.*, moving forward, turning left, moving backward, and turning right. The agent's ground truth one-step action is determined by the calculated orientation adjustment.

## C  Model Details

### C.1  MSR3D

#### C.1.1  Overall Structure

The overview structure of the MSR3D is illustrated in Fig. 19. This model is adapted from LEO [28], and further extended to accommodate text-image interleaved inputs. The tokenization for different modalities is stated as follows:

- **Text.** For texts in the instructions (*i.e.*, system messages, situation description, text in multimodal instruction, and response), we use SentencePiece tokenizer [33] to encode them with 32k subwords.

```
object_name = sample["object_name"]
end_situation = sample["end_situation"]
examples = " Given "Obejct description":   "relations": [ "to the left of table-35", "in front of
shelf-56", "placed on floor-9" ], "attributes":   "color": "white", "3D shape": "rectangular prism",
"material": "metal and plastic", "usage": "preserving food at low temperatures", "texture": "smooth",
"structure": "upright, single-door", "state": "closed and functioning" "object_name": "refrigerator",
You can generate results like "I want to open the white metal refrigerator.", "I want to close the white
refrigerator in front of the shelf." "I want to pick some food from the white refrigerator to the left of
table." "

prompt = f"""
Please generate a sentence about doing something with the object {object_name} based on the
following information: Obejct attributes and spatial relationship descriptions are given in the following
json: {end_situation}, you can select part of the given information to generate the sentence. The
generated sentence should be natural. here is an example:
Object descriptions is {examples} You can generate results like result: "I want to open the white metal
refrigerator.", result: "I want to close the white refrigerator in front of the shelf." result: "I want to
pick some food from the white refrigerator to the left of table." The generated results should be one
sentence in the follow format like: result: ".
"""

messages = [{"role": "user", "content": [{"type": "text", "text": prompt}]
```

Figure 18: Prompt messages to generate the goal action for MSNN.

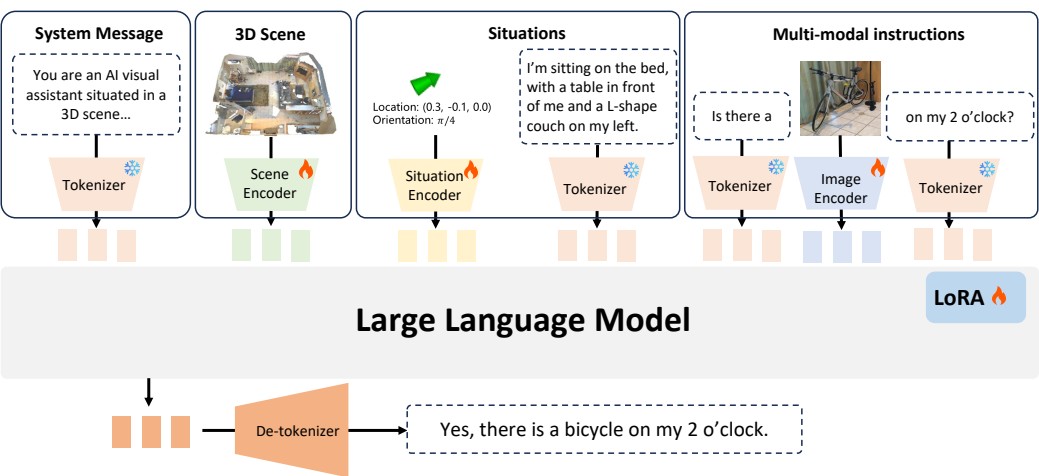

Figure 19: An overview of the MSR3D model.

- **3D Scene.** For 3D scene point clouds, we use object-centric representations as tokens for LLMs. Object proposals are extracted over a Mask3D-based 3D detection model, and the point clouds of the objects are fed into a pretrained PointNet++ network [48] to extract the object features. We then adopt a spatial transformer module [11] to model the spatial relationships between object proposals. All the output object features are sequenced and used as the scene tokens.
- **Situation.** The agent's location and orientation are utilized as a normalization for all the objects in the scene, *i.e.*, we rotate and translate all the objects in the scene to ensure the agent faces the same direction (positive direction along the x axis) at the same location (the origin point). Comparison between the proposed situation modeling method with other commonly used modeling strategies is provided in Tab. 17.
- **2D image.** For 2D images in the multi-modal instructions, we use a pretrained OpenCLIP ConvNeXt [39] model as the image encoder to get image token embeddings. A projection layer is applied to connect the dimensions.

Table 15: Hyperparameters for the training.

| Hyperparameter | Value |
|---|---|
| Optimizer | AdamW |
| Weight Decay | 0.05 |
| betas | [0.9, 0.999] |
| Learning Rate | $3 \times 10^{-5}$ |
| Warmup Steps | 400 |
| Type of GPUs | NVIDIA A100 |
| Number of GPUs | 4 |
| Accumulate Gradient Batches | 5 |
| Batch Size/GPU (total) | 4 (80) |
| gradient norm | 5.0 |
| epochs | 10 |

Table 16: Hyperparameters for inference.

| Hyperparameter | Value |
|---|---|
| Number of beams | 5 |
| maximum output length | 256 |
| repetition penalty | 3.0 |
| length penalty | 1.0 |

### C.1.2 Model Training

We train MSR3D in a GPT-style auto-aggressive prefix language modeling fashion, where the prefix spans from *system message* to *multi-modal instruction* and the target sequence to learn is *response*. We choose Vicuna-7B [16] as the base LLM to process the token sequence. In order to preserve the rich knowledge and strong reasoning capability of LLM, we use LoRA [25] to tune the LLM by introducing additional tunable parameters to the original LLM. We optimize the parameters $\theta$ of MSR3D in a prefix language modeling fashion, and the loss function for a batch $\mathcal{B}$ of the sequence $s$ is formulated as:

$$\mathcal{L}(\theta, \mathcal{B}) = -\sum_{b=1}^{|\mathcal{B}|} \sum_{t=1}^{T} \log p_\theta \left( s_{\text{pred}}^{(b,t)} | s_{\text{pred}}^{(b,<t)}, s_{\text{prefix}}^{(b,1)}, ..., s_{\text{prefix}}^{(b,L)} \right), \tag{1}$$

where $s_{\text{prefix}}$ is the prefix tokens and $s_{\text{pred}}$ denote the predicted tokens. For pre-training on MSQA and fine-tuning on downstream tasks, we freeze the parameters of LLM, PointNet++ object encoder, and tune the spatial transformer module, image encoder, image projection layer and LoRA parameters. During inference, we use beam search to generate the predicted texts.

The training consists of two stages: pretraining and task-specific finetuning. For both stages, we finetune the spatial Transformer and image encoder while freezing the point cloud encoder (PointNet++ [48]). We employ LoRA [25] to efficiently finetune the LLM (Vicuna-7B [16]), with rank = 16, $\alpha = 16$, and dropout disabled. The LoRA parameters are applied to all the projection matrices in the LLM. We list the hyperparameters of the two-stage training and inference as follows:

### C.1.3 Choice of Situation Modeling

We have detailed tested several situation modeling methods for MSR3D. The methods we have tested are stated as follows:

- **as object** The agent is treated as a special object in the scene with blank object features, and the situation of the agent is directly encoded using the scene encoder.
- **as embedding** The location and orientation of the agent are encoded as a special position embedding for all the other objects in the scene. The position embedding is computed by a projection layer form the original location and orientation to the object feature dimension.

Table 17: Comparision between different situation modeling methods on SQA3D dataset. Ground truth object proposals are used for testing. The refined exact match accuracy (EM-refined) [28] is used as the metric for evaluation.

| Situation Modeling Method | EM-refined |
|---------------------------|------------|
| as object                 | 51.92      |
| as embedding              | 50.72      |
| as cross attention        | 51.21      |
| as transformation         | **52.94**  |

- **as cross attention** The agent's situation is encoded as a condition for the scene encoder. We add a cross-attention layer for the situation modeling with the situation as query and the object features as key and value.
- **as transformation** The location and orientation of the agent are used as a normalization for all the objects. We rotate and translate all the objects in the scene to ensure the agent faces the same direction (positive direction along the x axis) at the same location (the origin point).

We tested the above modeling methods on SQA3D based on MSR3D, all the models are trained from scratch, and the results are given in Tab. 17. The results indicate that "as transformation" achieves the best performance, thus we choose this strategy for MSR3D.

### C.1.4 Prompt Details

The first part of the prompt is the system message, which is the same for pre-training and all downstream tasks. It is stated as follows:

> You are an AI visual assistant situated in a 3D scene. You can perceive (1) an ego-view image (accessible when necessary) and (2) the objects (including yourself) in the scene (always accessible). You should properly respond to the USER's instructions according to the given visual information. The USER's instructions may contain object-level information from images.

We use an object-centric 3D scene encoder to encode the 3D scene. We add a common sentence to the beginning of 3D scene tokens, shown as follows:

> Objects (including you) in the scene:

For the situation prompt, we add a common sentence before the situation description, shown as follows:

> You are at a selected location in the 3D scene.

When fine-tuning the model on the MSNN task, the instruction prompt we use is given as follows:

> {GOAL_ACTION}, what action should I take next step?

### C.1.5 Scene Encoder Details

We employ a frozen PointNet++ [48], pre-trained on the ScanNet [17] dataset with object classification task, to encode the objects present in the scene. For each object, we sample 1024 points, following the approach outlined in [11].

To capture the spatial relationships between different objects, we utilize a Spatial Transformer [11] module, a modified transformer architecture that explicitly encodes the spatial relations between object pairs. Specifically, consider the vanilla self-attention mechanism [59], which operates on a feature matrix $X \in \mathbf{R}^{N \times d}$, where $N$ represents the number of tokens and $d$ is the feature dimension. The self-attention mechanism first computes $Q = XW_Q$, $K = XW_K$, and $V = XW_V$ from

$X$ using learnable projection matrices $W_Q, W_K, W_V \in \mathbf{R}^{d \times d_h}$, where $d_h$ stands for the output feature dimension. Subsequently, the attention weight matrix is calculated as $(\omega_{ij}^o)_{N \times N} = \Omega^o = softmax(\frac{QK^T}{\sqrt{d_h}})$ and used to reweight $\Omega^o V$.

The intuition behind the Spatial Transformer is to rescale the elements $\omega_{ij}^o$ in the weight matrix $\Omega^o$ based on spatial information. In the object-centric reasoning setting, the input feature matrix is $O \in \mathbf{R}^{N \times d}$. For an object pair $(O_i, O_j)$ with geometric centers $c_i$ and $c_j$, the Spatial Transformer [11] computes the Euclidean distance $d_{ij} = ||c_i - c_j||2$ and the horizontal and vertical angles $\theta_h, \theta_v$ of the line connecting $c_i$ and $c_j$. The spatial feature between the two objects $(O_i, O_j)$ is a 5-dimensional vector $f_{ij} = [d_{ij}, \sin(\theta_h), \cos(\theta_h), \sin(\theta_v), \cos(\theta_v)]$. To combine this feature with the objects, the spatial attention computes $\omega_{ij}^s = g_i f_{ij}$, where $g_i = W_S^T o_i$ is a 5-dimensional vector. The spatial attention then reweights the original self-attention weight matrix as

$$\omega_{ij} = \frac{\sigma(\omega_{ij}^s) \exp(\omega_{ij}^o)}{\sum_{l=1}^N \sigma(\omega_{il}^s) \exp(\omega_{il}^o)}.$$

For more details, readers can refer to [11]. We employ a three-layer Spatial Transformer with 8 heads to process the object-centric features produced by PointNet++ and output object tokens. For other settings, we follow all the default hyperparameters in [11].

### C.2 Zero-shot Models

#### C.2.1 Prompts for MSQA

The prompts for MSQA using GPT-4o are stated in Fig. 20.

We replace all the images with the corresponding class labels for prompting GPT-3.5, and the prompt messages are stated in Fig. 21.

#### C.2.2 Prompts for MSNN

The prompts for MSNN using GPT-4o are stated in Fig. 22.

We also replace all the images with the corresponding class labels for prompting GPT-3.5, and the prompt messages are stated in Fig. 23.

## D Additional Experiments and Analysis

### D.1 Scaling Effect on MSNN

We also reveal the impact of scaling on the MSNN task by training MSR3D with different MSQA scales. Results are shown in Fig. 24. Observably, the performance improves as the MSQA scale increases, suggesting the effectiveness and scalability of MSQA.

### D.2 More Qualitative Results and Failure Cases in MSQA

We provide more qualitative examples and failure cases in Fig. 25 and Fig. 26. The results manifest 1) GPT-4o struggles in spatial reasoning even provided with accurate object coordinates and sizes; 2) current models show insufficient abilities in perception and reasoning when handling the situated reasoning task in MSQA.

### D.3 Additional analyses for situation component

In the analyses of situation component in Tab. 2, the difference in Exist. and Spatial is minor. We conjecture these two domains contain many questions that are agnostic to situation. Therefore, we conduct additional experiments on the subsets where question answering is highly dependent on the cues from situation. Specifically, we consider two addtional settings regarding such a hypothesis and present the analyses as follows.

**Exist.** For Exist., we filter the questions querying in-the-wild objects (e.g., car, elephant) since these questions (e.g., "Is there a car on my right?") can be answered without understanding the situation and scene.

```
scene_format = "inst_name: [x, y, z], [h, w, d], color, 3D shape, material, usage, texture, structure,
state;"
answer_format = "Answer:"
scene_info_prompt = sample["scene_info_json"]
location = sample["location"]
orientation = sample["orientation"]
question = sample["question"]
situation = sample["situation"]
image_order = sample["image_order"]
image_list = [encode_image(image_path) for image_path in sample["image_paths"]]

prompt = f"""
You are an AI visual assistant situated in a 3D scene. You can perceive the objects (including yourself)
in the scene. The scene representation is given in a dict format such as {scene_format}. All object
instances in this room are given, along with their center point position and size. The center points are
represent by a 3D coordinate (x, y, z) with units of meters, and the bounding boxes are represent by
(h, w, d) with units of meters along the x, y and z coordinate. The attribute of the objects are also
provided in the scene representation. The objects in the scene are: {scene_info_prompt} You are an
agent in the three-dimensional environment. Your situation is {situation}. Your location {location} is
given by a 3D coordinate (x, y, z) with units of meters. and you are facing the direction in x-y plane
with an angle of {face_angle}.
USER : {question}
You should properly respond to the USER's instruction according to the given information. You should
directly response to the question. The output answer should follow the format like {answer_format}.
There are some objects represented by image in this situation and question prompt. The image is given
as bellow to replace the format like "<>" in this prompt. Image order is : {image_order}
ASSISTANT:
"""

content_list = [{"type": "text", "text": prompt}]
content_list.extend([{"type": "image_url",
"image_url": { "url": f"data:image/jpeg;base64,{image_base}"}} for image_base in image_list])
messages = [{"role": "user", "content": content_list]
```

Figure 20: Prompt messages for GPT-4o on MSQA task.

Table 18: Analysis for object existence.

| FT model | Exist. | Exist. @ w/o in-the-wild objects |
|---|---|---|
| T | 86.28 | 84.62 |
| T+I | 85.51 | 82.03 |
| Δ | -0.77 | -2.59 |

Table 19: Analysis for spatial relationship.

| FT model | Spatial | Spatial @ directional answer | Spatial @ object refer |
|---|---|---|---|
| w/ situation | 42.79 | 15.08 | 43.93 |
| w/o situatio | 42.66 | 14.28 | 42.47 |
| Δ | -0.13 | -0.80 | -1.46 |

The results in Tab. 18 support our hypothesis since the impact of situation component is amplified after eliminating the questions regarding in-the-wild objects.

**Spatial.** For this type of question, we recognize two scenarios where question answering is highly related to situation: (1) directional answer, where GT answer contains some directional phrases such as "left" and "behind"; and (2) object refer, where the question focuses on spatial relations between objects.

The results in Tab. 19 show notable differences in the two above scenarios, also supporting our hypothesis, i.e., situation matters in those situation-dependent questions.

scene_format = "inst_name: [x, y, z], [h, w, d], color, 3D shape, material, usage, texture, structure, state;"
answer_format = "Answer:"
scene_info_prompt = sample["scene_info_json"]
location = sample["location"]
orientation = sample["orientation"]
question = sample["question"]
situation = sample["situation"]

prompt = f"""
You are an AI visual assistant situated in a 3D scene. You can perceive the objects (including yourself) in the scene. The scene representation is given in a dict format such as {scene_format}. All object instances in this room are given, along with their center point position and size. The center points are represent by a 3D coordinate (x, y, z) with units of meters, and the bounding boxes are represent by (h, w, d) with units of meters along the x, y and z coordinate. The attribute of the objects are also provided in the scene representation. The objects in the scene are: {scene_info_prompt} You are an agent in the three-dimensional environment. Your situation is {situation}. Your location {location} is given by a 3D coordinate (x, y, z) with units of meters. and you are facing the direction in x-y plane with an angle of {face_angle}.
USER : {question}
You should properly respond to the USER's instruction according to the given information. You should directly response to the question. The output answer should follow the format like {answer_format}.
ASSISTANT:
"""

messages = [{"role": "user", "content": [{"type": "text", "text": prompt}]}]

Figure 21: Prompt messages for GPT-3.5 on MSQA task.

```
scene_format = "inst_name: [x, y, z], [h, w, d], color, 3D shape, material, usage, texture, structure,
state;"
answer_format = "Answer: the number of the action you choose"
scene_info_prompt = sample["scene_info_json"]
location = sample["location"]
orientation = sample["orientation"]
question = sample["goal_action"] + " What action should I take next?"
situation = sample["situation"]
choices = "0: move forward; 1: turn left; 2: move backward; 3: turn right;"
image_order = sample["image_order"]
image_list = [encode_image(image_path) for image_path in sample["image_paths"]]

prompt = f"""
You are an AI visual assistant situated in a 3D scene. You can perceive the objects (including yourself)
in the scene. The scene representation is given in a dict format such as {scene_format}. All object
instances in this room are given, along with their center point position and size. The center points are
represent by a 3D coordinate (x, y, z) with units of meters, and the bounding boxes are represent by
(h, w, d) with units of meters along the x, y and z coordinate. The attribute of the objects are also
provided in the scene representation. The objects in the scene are: {scene_info_prompt} You are an
agent in the three-dimensional environment. Your situation is {situation}. Your location {location} is
given by a 3D coordinate (x, y, z) with units of meters. and you are facing the direction in x-y plane
with an angle of {face_angle}.
USER : {question}
You should properly respond to the USER's instruction according to the given information. You should
directly response to the question. The output answer should follow the format like {answer_format}.
There are some objects represented by image in this situation and question prompt. The image is given
as bellow to replace the format like "<>" in this prompt. Image order is : {image_order}
You should properly and directly choose one answer from the following options and return the index
of the answer. The options are as follows: {choices}  And your answer should be a single number in a
format {answer_format}.  ASSISTANT:
"""

content_list = [{"type": "text", "text": prompt}]
content_list.extend([{"type": "image_url",
"image_url": { "url": f"data:image/jpeg;base64,{image_base}"}} for image_base in image_list])
messages = [{"role": "user", "content": content_list}]
```

Figure 22: Prompt messages for GPT-4o on MSNN task.

```
scene_format = "inst_name: [x, y, z], [h, w, d], color, 3D shape, material, usage, texture, structure,
state;"
answer_format = "Answer: the number of the action you choose"
scene_info_prompt = sample["scene_info_json"]
location = sample["location"]
orientation = sample["orientation"]
question = sample["goal_action"] + " What action should I take next?"
situation = sample["situation"]
choices = "0: move forward; 1: turn left; 2: move backward; 3: turn right;"

prompt = f"""
You are an AI visual assistant situated in a 3D scene. You can perceive the objects (including yourself)
in the scene. The scene representation is given in a dict format such as {scene_format}. All object
instances in this room are given, along with their center point position and size. The center points are
represent by a 3D coordinate (x, y, z) with units of meters, and the bounding boxes are represent by
(h, w, d) with units of meters along the x, y and z coordinate. The attribute of the objects are also
provided in the scene representation. The objects in the scene are: {scene_info_prompt} You are an
agent in the three-dimensional environment. Your situation is {situation}. Your location {location} is
given by a 3D coordinate (x, y, z) with units of meters. and you are facing the direction in x-y plane
with an angle of {face_angle}.
USER : {question}
You should properly respond to the USER's instruction according to the given information. You
should directly response to the question. You should properly and directly choose one answer from the
following options and return the index of the answer. The options are as follows: {choices}  And your
answer should be a single number in a format {answer_format}.
ASSISTANT:
"""

messages = [{"role": "user", "content": [{"type": "text", "text": prompt}]]]
```

Figure 23: Prompt messages for GPT-3.5 on MSNN task.

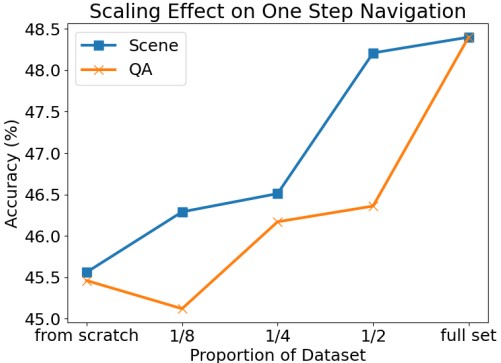

Figure 24: Scaling effect on MSNN for MSR3D pretrained on different MSQA scales.

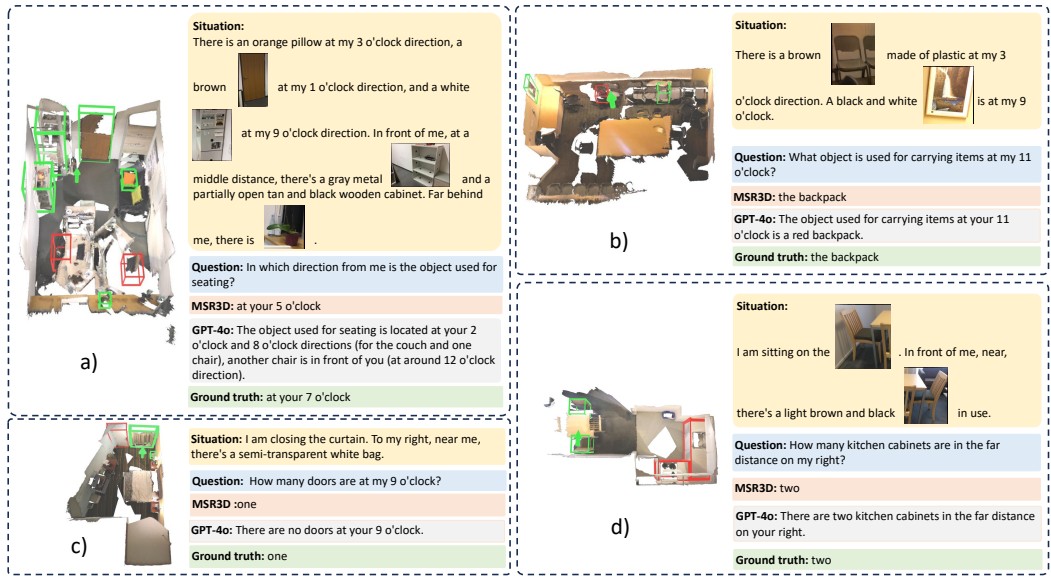

Figure 25: Additional qualitative results in MSQA. a) MSR3D's answer is reasonable according to the scene, while the answer of GPT-4o is unreasonable. b) The answers of MSR3D and GPT-4o are both correct. c) The answer of GPT-4o is incorrect. d) The answers of both models are exactly correct.

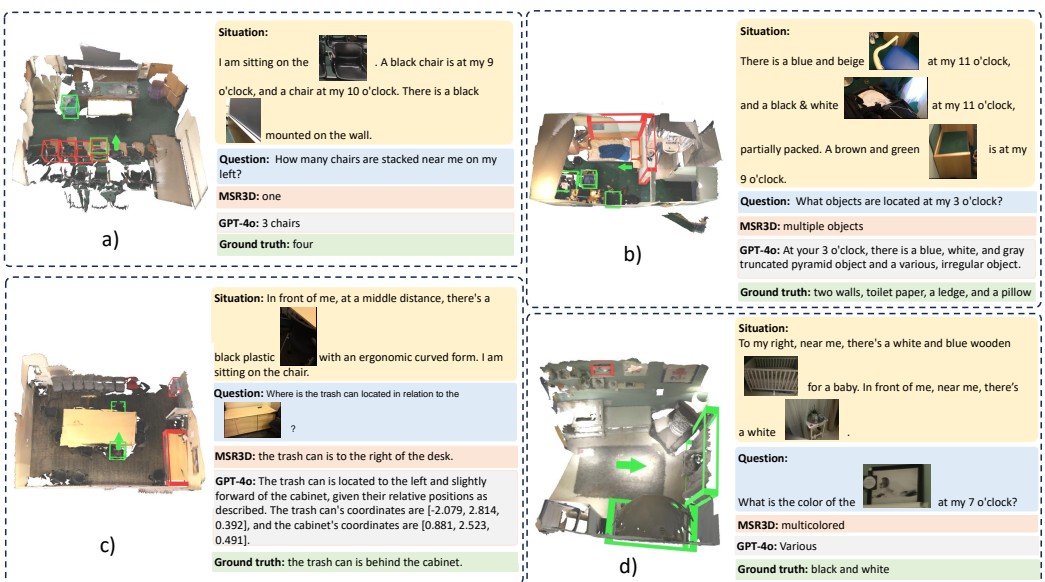

Figure 26: Failure cases in MSQA. a) GPT-4o's answer is close to ground truth, while MSR3D's is far from the ground truth. b) Both models cannot provide the accurate names and quantity of the queried objects. c) MSR3D incorrectly answers the spatial relationship. GPT-4o only provides the coordinates of objects without the spatial relationship. d) Both models cannot provide accurate colors of the picture.

