# OpenReview forum: "Multi-modal Situated Reasoning in 3D Scenes"
_NeurIPS.cc/2024/Datasets_and_Benchmarks_Track — NeurIPS 2024 Track Datasets and Benchmarks Poster_

### Official Review · Reviewer_cvGg · 2024-07-21
**This work presents a new multimodal dataset and defines new navigation tasks.**

**Rating:** 7
**Confidence:** 4
**Correctness:** Correct
**Clarity:** yes

**Review:**

The authors created a data annotation pipeline to develop a new multimodal indoor dataset named MSQA, which aims to overcome the constraints of single-modal input. They introduced the MSNN benchmark for this dataset to evaluate how well models comprehend transitions between actions and situations. Additionally, they established a new baseline by making adjustments to the LEO model.

Pros:

Introduces a comprehensive and diverse 3D scene understanding dataset, MSQA, that addresses existing gaps.

Integrates interwoven multimodal input settings and MSNN tasks, better simulating real-world applications.

Ensures data quality through LLM-assisted generation and manual review.

Cons:
The experimental section by the authors is not comprehensive enough and does not highlight the effectiveness of the MSQA dataset's "T+I" mode in enhancing LLM's spatial perception abilities.

The baseline improvements made on LEO are lower than LEO in the basic metrics of Count and Exist.

**Strengths:**

see above

**Additional Feedback:**

none

**Documentation:**

Complete

**Limitations:**

Exist

**Opportunities For Improvement:**

According to Table 2, the spatial score is highest with text-only input, which fails to prove whether the text+image input is effective or redundant. Adding images does not seem to enhance spatial understanding and reasoning performance.
Did the authors compare the accuracy of the FT model without the situation? It seems inevitable that QA scores would improve after FT, but it does not prove the effectiveness of your situation component.

**Relation To Prior Work:**

Innovative compared to previous datasets

**Summary And Contributions:**

Based on the existing dataset, the authors designed a data annotation pipeline to introduce a new multimodal indoor dataset called MSQA, aiming to address the limitations of single-modal input. They proposed the MSNN benchmark on this dataset to evaluate the model's understanding of transitions between actions and situations. By comparing with existing large multimodal models, they verified that pre-training on MSQA enhances the model's reasoning performance in 3D scenes. Additionally, they introduced a new baseline by modifying the LEO model.

---

> ### Author Rebuttal · Authors · 2024-08-17
>
> Thanks for your valuable feedback. We will address your concerns below.
>
> > Cons:
> > The experimental section by the authors is not comprehensive enough and does not highlight the effectiveness of the MSQA dataset's "T+I" mode in enhancing LLM's spatial perception abilities.
> > The baseline improvements made on LEO are lower than LEO in the basic metrics of Count and Exist.
>
> > Opportunities For Improvement:
> > According to Table 2, the spatial score is highest with text-only input, which fails to prove whether the text+image input is effective or redundant. Adding images does not seem to enhance spatial understanding and reasoning performance. Did the authors compare the accuracy of the FT model without the situation? It seems inevitable that QA scores would improve after FT, but it does not prove the effectiveness of your situation component.
>
> We summarize your feedback into three points and will address them one by one:
>     - *Why "T+I" is worse than "T" in spatial reasoning.*
>     - *Why MSR3D is worse than the baseline LEO in object existence and counting.*
>     - *How much does the situation matter?*
>
> ---
> ***Why "T+I" is worse than "T" in spatial reasoning***
>
> Compared with the "T" setting, the incorporated images in the "T+I" setting can appear in both *situation* and *question*. To investigate the subtle performance difference between "T" and "T+I", we extract two subsets from the test data by making the images only appear in either *situation* or *question*. Then we test the models ("T" and "T+I") on the two subsets, respectively. The results are as below.
>
> |Model|situation w/ image, question w/o image|situation w/o image, question w/ image|
> | -------- | -------- |-------- |
> |T|55.54|56.41|
> |T+I|56.48|43.58|
> |$\Delta$|0.94|-12.83|
>
> We observe that "T+I" suffers a significant drop in the subset where the images only appear in *question*. This indicates that **incorporating images in *question* makes it more challenging to answer** probably because identifying the queried objects from images is harder than from text. Our finding means **the "T+I" model is not weaker**. Instead, **the drop stems from a harder evaluation setting**.
>
> ---
> ***Why MSR3D is worse than the baseline LEO in object existence and counting***
>
> Despite the minor drops in Count. and Exist., MSR3D still shows a **better overall performance (LEO "T" 55.86 vs. MSR3D "T" 56.48)**. In particular, the implementation of MSR3D **focuses more on spatial reasoning and navigation, wherein we observe significant improvements (especially +8.56 for MSNN)**. More sophisticated designs in terms of model would be considered in our future work.
>
> ---
> ***How much does the situation matter***
>
> To reveal the effectiveness of situation for FT models, we add an FT model with the entire situation component removed while retaining the 3D scene and question as input. The detailed results are as follows:
>
> | FT setting | Count. |Exist. | Attr. |Spatial|Navi.|Others|Overall|
> | -------- | -------- |--------| -------- |--------|--------|--------|--------|
> | w/ situation     | **33.46**    |**86.28**   | **50.88**    |**42.79**|**62.56** |**73.31**|**54.13**|
> | w/o situation     | 30.78 |85.51| 45.35 |42.66| 52.97 |71.00 |51.20 |
>
> The results show a significant drop in performance after removing the situation component, demonstrating the effectiveness of our situation component. In particular, the drop in questions related to navigation is more salient, which echoes the evaluation results in MSNN and highlights the importance of the situation component.
>
> We also notice the minor difference in Exist. and Spatial. We conjecture these two domains contain many questions that are agnostic to situation. Therefore, we conduct additional experiments on the subsets where question answering is highly dependent on the cues from situation. Specifically, we consider two additional settings regarding such a hypothesis and present the analyses as follows.
>
> **Exist.** For Exist., we filter the questions querying in-the-wild objects (*e.g.*, car, elephant) since these questions (*e.g.*, "Is there a car on my right?") can be answered without understanding the situation and scene.
>
> |FT model|Exist.|Exist. @ w/o in-the-wild objects|
> | -------- | -------- |--------|
> |w/ situation|86.28|84.62|
> |w/o situation|85.51|82.03|
> |$\Delta$|-0.77|-2.59|
>
> The results support our hypothesis since the impact of the situation component is amplified after eliminating the questions regarding in-the-wild objects.
>
> **Spatial.** For this type of question, we recognize two scenarios where question answering is highly related to situation: (1) directional answer, where GT answer contains some directional phrases such as "left" and "behind"; and (2) object refer, where the question focuses on spatial relations between objects.
>
> |FT model|Spatial|Spatial @ directional answer|Spatial @ object refer|
> | -------- | -------- |--------|--------|
> |w/ situation|42.79|15.08|43.93|
> |w/o situation|42.66|14.28|42.47|
> |$\Delta$|-0.13|-0.80|-1.46|
>
> The results show notable differences in the two above scenarios, also supporting our hypothesis, *i.e.*, situation matters in those situation-dependent questions.
>
> ---
>
> In summary, we demonstrate the effectiveness of the situation component, particularly in tasks such as navigation. We will incorporate the additional experiments and detailed analyses into our revised version. We hope our results can address your questions and would appreciate it if you could consider raising the score.

---

> ### Author Response · Authors · 2024-08-23
>
> Dear Reviewer cvGg,
>
> Thanks again for your valuable feedback! We have conducted additional experiments to address your concerns. Please feel free to let us know if anything further requires clarification.
>
> Best,
>
> The Authors

---

### Official Review · Reviewer_RR5c · 2024-07-23
**Review of 695**

**Rating:** 6
**Confidence:** 4
**Correctness:** Yes.
**Clarity:** Good.

**Review:**

Strengths

1.	The paper concentrates on the core challenge of embodied AI, situation awareness in the multimodal local context with the environment, which plays a pivotal role in agents’ understanding and interactions with the 3D physical environment.

2.	Then they develop an automated data-curated pipeline, which effectively solves the limitation of the existing method which has an expensive data collection process. In this way, they collect about 251 K situated QA pairs, covering more complex scenarios and object modality than previous works.

3.	The two benchmarks MSQA and MSNN are devised based on curated data, which expands the scopes of existing situation QA tasks and simplifies the traditional multi-step embodied navigation to a single-step setting.

4. They execute extensive experiments to investigate the capability of existing vision-language models on these tasks. Simultaneously, they testify to the potential of VLM in fine-tuning these tasks, which provided a broad insight for training more powerful 3D visual-language models.

Weakness

1.	Typos, such as on line 130 ‘OUr’.

2.	It deserves to be noted that involving GPT-3.5 in Situation Sampling and QA Pairs generation, produces a curious about whether the surrounding object description and situated question-answer pairs produced by GPT-3.5 have negligible differences compared with GPT-4. It would be better if could give some evidence to support that GPT-3.5 has the same capability as GPT-4 in this case.

3.	The method involves the GPT-based model in the loop, i.e. GPT-3.5 and GPT-4, and the experiments also evaluate these models’ zero-shot capability. There is a lack of evidence about the bias or preference induced by the GPT-based model. Hence, it would be better if could evaluate the zero-shot capability of other close-sourced models, such as Google’s Gemini.

4.	In the experiment of instruction tuning, including two different settings, replacing the input image with their corresponding category label and accommodating the interleaved multi-modal input setting. It lacks a description of the 3D scenes, but it is curious about the performance difference between the textual description of 3D scenes and real 3D scenes input.

**Strengths:**

Please refer to the pros above.

**Additional Feedback:**

Please fix the typos and provide more discussions for the proposed issues.

**Documentation:**

Yes, in the suppl.

**Limitations:**

Yes.

**Opportunities For Improvement:**

Please refer to the weakness.

**Relation To Prior Work:**

Clear.

**Summary And Contributions:**

This paper investigates the situation awareness of 3D embodied agents. To solve the limitation in existing datasets and benchmarking, simulated environments have pool quality of situation data according to constrained diversity and complexity of synthetic scenes and real-world scenes involving expensive data collection cost, this paper proposes a Multi-modal Situated Question Answer (MSQA) dataset that is collected from the 3D scene graph and Visual-Language Model (VLM) in 3D real-world scenes. The dataset covers 9 distinct scenes and involves complex task and object modality in 3D scenes. To solve the ambiguity of single-modality, they include an interleaved multi-modal input setting with text, image, and point cloud. Additionally, they propose Multi-modal Situation Question Answer (MSQA) and Multi-modal Next-step Navigation (MSNN) benchmarks to comprehensively investigate the model’s capability in embodied reasoning and navigation. Expanded results demonstrate that existing VLMs have pool performance in these tasks and the potential of using these datasets to develop a more powerful situation reasoning model.

---

> ### Author Rebuttal · Authors · 2024-08-17
>
> Thanks for your valuable feedback. We will address your concerns as below.
>
> > Weakness
> > 1. Typos, such as on line 130 ‘OUr’.
>
> Thanks for pointing out the typo. We will correct the typo in the revised version.
>
> ---
>
> > 2. It deserves to be noted that involving GPT-3.5 in Situation Sampling and QA Pairs generation, produces a curious about whether the surrounding object description and situated question-answer pairs produced by GPT-3.5 have negligible differences compared with GPT-4. It would be better if could give some evidence to support that GPT-3.5 has the same capability as GPT-4 in this case.
>
> To compare the QA pairs generated by GPT-4o (the most advanced version) and GPT-3.5, we conducted a human study on the raw data generated by each LLM. The evaluation assesses correctness from three aspects:
>
> - **Correctness of chain of thought**: whether the object chain of thought is coherent and relevant to the question.
> - **Correctness of answer**: whether the generated answer is accurate.
> - **Correctness of format**: whether the generated data adheres to the expected format, *e.g.*, <table-4-M>.
>
> | LLM | Acc. @ COT |Acc. @ answer| Acc. @ format |# QA pairs|
> | -------- | -------- |--------| -------- |--------|
> | GPT-3.5     | 88.23    |**88.23**    | **96.88**    |51|
> | GPT-4o     | **92.45**    |86.97   | 62.5   |53|
>
> The results indicate comparable performances between GPT-3.5 and GPT-4o in terms of the correctness of chain-of-thought reasoning and final answers. However, GPT-3.5 significantly outperforms GPT-4o in terms of format correctness. We observed that, with the same system prompts and demonstrations as GPT-3.5, GPT-4o occasionally produces data in incorrect format, such as <table-4-N> or <chair-6>. These anomalies result in unexpected errors during the parsing of placeholders, which ultimately affects the data processing workflow. Therefore, we opted to use GPT-3.5 for data generation.
>
> All relevant details, including the principles behind scoring, as well as examples of generated data and scene graphs from the evaluation process, can be found in the [human study for GPT models in data generation](https://docs.google.com/document/d/16UVB-mCSt6ZFynABlFGjVB6fpEuJx3c2AjazIhtFAps/edit).
>
> ---
>
> > 3. The method involves the GPT-based model in the loop, i.e. GPT-3.5 and GPT-4, and the experiments also evaluate these models’ zero-shot capability. There is a lack of evidence about the bias or preference induced by the GPT-based model. Hence, it would be better if could evaluate the zero-shot capability of other close-sourced models, such as Google’s Gemini.
>
> Good suggestion! To eliminate the potential bias from the GPT family of models, we tested another state-of-the-art, closed-source model, **Claude-3.5-Sonnet-20240620**. Due to regional policy restrictions, we were unable to access Gemini's API, so we opted for the latest version of Claude-3.5-Sonnet as an alternative, which has demonstrated comparable performances to GPT-4o in many scenarios.
>
> | FT setting | Count. |Exist. | Attr. |Spatial|Navi.|Others|Overall|
> | -------- | -------- |--------| -------- |--------|--------|--------|--------|
> | GPT-4o    | 31.20 |**71.41** |**75.21** |**31.50**| 36.67| **88.03**| 49.68|
> | Claude-3.5-Sonnet    | **32.57**| 66.28 |69.88 |30.10| **45.48** |83.61 |**49.73**|
>
> The results indicate that the overall performance of Claude-3.5-Sonnet on the MSQA benchmark is very close to that of GPT-4o. For specific sub-tasks, Claude-3.5-Sonnet is better at *navigation* and *counting* yet worse at other sub-tasks compared to GPT-4o. We believe this additional experiment could eliminate the potential bias within GPT family and consolidate the evaluation results of current large VLMs' zero-shot performances on MSQA.

---

> > ### Comment · Reviewer_RR5c · 2024-08-24
> > **post-rebuttal**
> >
> > Thanks for your detailed response. You’ve addressed my main concern.
> >
> > I still have some confusion regarding Answer 1 in the rebuttal: ‘Given this limitation, each caption typically consists of 4-5 sentences, which may result in some objects from the 3D scenes being omitted.’ Could you clarify the strategy used to compress the context in each caption?
> >
> > Regarding the comparison between GPT-3.5 and GPT-4.0, does the strict requirement for output formatting constrain GPT-4.0’s performance? If so, it is recommended to maximize the model’s capabilities and explore new strategies for format conversion.

---

> > > ### Author Response · Authors · 2024-08-25
> > > **Further clarification**
> > >
> > > > I still have some confusion regarding Answer 1 in the rebuttal: ‘Given this limitation, each caption typically consists of 4-5 sentences, which may result in some objects from the 3D scenes being omitted.’ Could you clarify the strategy used to compress the context in each caption?
> > >
> > > We utilize ChatGPT to select objects within the situated scene graphs and generate textual descriptions for them. During the post-processing phase, we assess the token length of each caption. If any caption exceeds the token limit, redundant sentences are removed to ensure compliance.
> > >
> > > > Regarding the comparison between GPT-3.5 and GPT-4.0, does the strict requirement for output formatting constrain GPT-4.0’s performance? If so, it is recommended to maximize the model’s capabilities and explore new strategies for format conversion.
> > >
> > > Thank you for your suggestions. We believe that further improvements in prompt design, refinement, and demonstration could indeed enhance GPT-4's performance in format standardization. Previous work indicates that GPT-4 may outperform GPT-3.5, particularly with the inclusion of bounding boxes in prompts. However, our current approach utilizes a meticulously designed scene graph generation and post-processing pipeline. This careful design ensures high data quality and reduces the complexity for the GPT model in inferring spatial relationships and generating accurate questions and answers. As a result, GPT-3.5 is sufficient for meeting the requirements of our current work.
> > >
> > > We hope our response can address your confusion and would appreciate it if you could consider raising the score.

---

> ### Author Rebuttal · Authors · 2024-08-17
>
> >4. In the experiment of instruction tuning, including two different settings, replacing the input image with their corresponding category label and accommodating the interleaved multi-modal input setting. It lacks a description of the 3D scenes, but it is curious about the performance difference between the textual description of 3D scenes and real 3D scenes input.
>
> Good suggestion! We conduct an additional experiment with solely textual descriptions as the input, *i.e.*, input images replaced by category labels as you suggested.
>
> **Implementation**
>     - Data. We utilized GPT-3.5 to derive textual descriptions of the 3D scenes based on the situated scene graphs. The situations used for generating the textual descriptions aligns with the original situations of QA pairs in MSQA. Examples of the textual descriptions of the 3D scenes can be found [here](https://docs.google.com/document/d/1VJzy5it-G47AN6IAxoKEOdqwIEVI3FtCxA9VohdRDcE/edit?usp=sharing).
>     - Model. We removed the scene tokens and adopted the textual descriptions as the only model input. Due to the maximum context length as per MSR3D’s setting (256 tokens), the textual descriptions may be truncated (*e.g.*, no more than 4~5 sentences) and thus some objects may be omitted.
>
> **Results**
>
> | Model input | Count. |Exist. | Attr. |Spatial|Navi.|Others|Overall|
> | -------- | -------- |--------| -------- |--------|--------|--------|--------|
> | Point cloud   | 33.46    |86.28   | **50.88**    |**42.79**|**62.56** |**73.31**|**54.13**|
> | Textual description    | **35.82** |**88.20**| 43.91 |35.52| 52.42 |73.10 |50.05|
>
> The results indicate that when provided with textual descriptions, the model’s performances significantly drop in terms of object attribute, spatial relation, and navigation. On the other hand, we observe higher performances in terms of object existence and counting. Furthermore, we delved into the questions about object existence and counting, and analyzed the performances according to the characteristics of ground truth.
>
> **Object existence: GT=yes *vs.* GT=no**
>
> |Model input|Exist.|Exist. @ GT=yes|Exist. @ GT=no|
> | -------- | -------- |--------|--------|
> |Point cloud|86.28|87.38|84.88|
> |Textual description|88.20|90.82|84.88|
> |$\Delta$|1.92|3.44|0|
>
> We found the performance improvement in terms of object existence almost comes from the questions where the GT is "yes". This implies that the model can more easily identify the existence of the queried object when it is explicitly depicted in the textual descriptions.
>
> **Counting: GT={1,2,3,4,5}**
>
> |Model input|Count.|Count. @ GT=1|Count. @ GT=2|Count. @ GT=3|Count. @ GT=4|Count. @ GT=5|
> | -------- | -------- |--------|--------|--------|--------|--------|
> |Point cloud|33.46|21.62|61.36|36.73|3.22|14.29
> |Textual description|35.82|32.43|82.95|10.20|0|0|
> |$\Delta$|2.36|10.81|21.59|-26.53|-3.22|-14.29|
>
> The results show: (1) Using textual descriptions as the input is better when GT ≤ 2, which is intuitive since the target objects are explicitly mentioned in textual descriptions. (2) Using textual descriptions as the input is significantly worse when GT ≥ 3, which probably stems from omitted objects due to context truncation. This indicates the difficulty of representing 3D scenes with textual descriptions, especially when the scenes become more complex and the object number increases.
>
> In summary, the results demonstrate that the 3D point cloud input serves as a more efficient representation for situated reasoning compared to textual descriptions. We will incorporate the additional experiments and detailed analyses into the revised version.

---

> ### Author Response · Authors · 2024-08-23
>
> Dear Reviewer RR5c,
>
> We have conducted additional experiments and analyses to address your concerns. We would appreciate it if you could review our responses and let us know if they adequately resolve the issues you raised. If there are any further clarifications needed or if we've overlooked any points, please don’t hesitate to let us know.
>
> Thank you once again for your valuable feedback.
>
> Best regards,
>
> The Authors

---

### Official Review · Reviewer_pZjR · 2024-07-24
**Review: MSQA**

**Rating:** 7
**Confidence:** 4
**Correctness:** Sound and correct.
**Clarity:** Paper is well-written.

**Review:**

Strengths

- The paper exhibits relatively good clarity and coherence in presenting its concepts. The accompanying figures serve as effective visual aids, significantly enhancing the comprehensibility of the discussed ideas.
- The paper presents a clear motivation for the research. The motivation behind an interleaved dataset for 3D multi-modal reasoning has a strong potential impact in the field of 3D computer vision and embodied AI.
- I love the concept of interleaved dataset in 3D multi-modal reasoning domain. This data format aligns closer with an embodied agent which navigates itself in a 3D scene with all of its perception methods along with language guidance. This is a missing block in the current literature and this benchmark marks a great step towards developing a more realistic 3D perception and reasoning generalist.

Weakness

- Visual prompting in situational understanding. I appreciate the effort of authors in proposing an interleaved dataset for situational understanding. However, the current benchmark seems still asking an agent to largely rely on language prompts to locate its own situation. I think this might be a little bit disconnected with the actual embodied scenario, where an agent relies heavily on its visual input to locate itself. For example, I can curate the following situational description: My front-facing camera is giving me this image <image of the camera>, what is my situation?
- The example above is also an “interleaved” input, except it puts more weights into the visual input rather than language. I believe this might align closer with the embodied AI setting.
- Scene data might be a bit limited and small. I understand MSQA is already a benchmark with the largest number of 3D scenes. However, I would like to hear authors' comments on the feasibility of scaling up the number of scenes even more -- for example, by leveraging large-scale synthetic 3D datasets available. Your proposed dataset curation pipeline is LLM-based and automatic. I think the paper will have an even bigger contribution if the number of the 3D scene can be scaled to 10K, much larger than the scale of ScanNet that everyone is using. Moreover, using synthetic dataset also makes it possible to generate more navigation trajectories easily.
- Missing related work. A seemingly very relevant work on 3D situational awareness [A] is missing in the current manuscript.

Writing Issues

- Formatting issues, when multiple citations are in the same bracket, they should be in monotonically increasing order, rather than random order. (For example, in line 109, 111)
- Figure 2 is referred to in the text before Figure 1, which is a small structural issue.

These points highlight areas for significant improvement in both the content and presentation of the research

References

[A] Man, Y., Gui, L. Y., Wang, Y. X. Situational Awareness Matters in 3D Vision Language Reasoning. In CVPR 2024.

**Strengths:**

The paper exhibits relatively good clarity and coherence in presenting its concepts. The accompanying figures serve as effective visual aids, significantly enhancing the comprehensibility of the discussed ideas. The paper presents a clear motivation for the research. The motivation behind an interleaved dataset for 3D multi-modal reasoning has a strong potential impact in the field of 3D computer vision and embodied AI.

**Additional Feedback:**

Here are a few questions related to weakness points.

1. Given the potential disconnect between your current benchmark and real-world embodied AI scenarios, how might you modify your dataset to place more emphasis on visual inputs for situational understanding? Have you considered creating scenarios where the agent must rely primarily on visual cues to determine its situation?
2. While MSQA offers a large number of 3D scenes, have you explored the possibility of significantly scaling up the dataset, perhaps to 10,000 scenes or more? What challenges do you foresee in leveraging large-scale synthetic 3D datasets to achieve this, and how might such an expansion enhance the contribution of your work?
3. Your LLM-based dataset curation pipeline seems well-suited for automation. Have you considered applying this method to generate a much larger dataset, potentially surpassing the scale of commonly used datasets like ScanNet? What benefits and challenges do you anticipate in such an endeavor?
4. The omission of the relevant work on 3D situational awareness [A] is notable. How does your work relate to or differ from this study, and how might incorporating a discussion of this paper enhance your literature review?

**Documentation:**

Sufficient details.

**Ethics:**

No ethical concerns.

**Limitations:**

The discussion on limitations about the paper is enough. The points raised in the weaknesses section further articulate limitations that should be considered.

**Opportunities For Improvement:**

See weaknesses.

**Relation To Prior Work:**

Clearly discussed.

**Summary And Contributions:**

This paper introduces MSQA, a large-scale multi-modal situated reasoning dataset, scalably collected leveraging 3D scene graphs and vision-language models (VLMs) across a diverse range of real-world 3D scenes. The proposed dataset innovatively introduces interleaved multiple input settings into the 3D VQA domain.  Experiments demonstrate the effectiveness of leveraging MSQA as a pre-training dataset for developing more powerful situated reasoning models, contributing to advancements in 3D scene understanding for embodied AI.

---

> ### Author Rebuttal · Authors · 2024-08-17
>
> Thanks for your valuable feedback. We will address your concerns as below.
>
> ---
>
> > Weaknesses
> > 1. Visual prompting in situational understanding. (...). For example, I can curate the following situational description: My front-facing camera is giving me this image \<image of the camera>, what is my situation?
>
> Good point! We agree that locating situation through visual cues is straightforward for embodied AI. Actually, this could be covered in our interleaved scene-image-text setting, which also requires the understanding of visual input such as egoview image. Extending the task with more focus on visual input (*e.g.*, egoview image) to tailor for embodied AI is really an interesting direction and can be explored in our future work. Anyway, we respectfully don't think this is a weakness. Instead, we think it an interesting and underexplored topic which is out of the scope of this paper.
>
> ---
>
> > 2. The example above is also an “interleaved” input, except it puts more weights into the visual input rather than language. I believe this might align closer with the embodied AI setting.
>
> The use of visual input is indeed well-suited for the embodied AI setting, particularly when assessing an agent's ability to locate itself within a scene. In our work, we aim to evaluate not only the agent's situational awareness but also its reasoning capabilities.
>
> ---
>
> > 3. Scene data might be a bit limited and small. I understand MSQA is already a benchmark with the largest number of 3D scenes. (...) Moreover, using synthetic dataset also makes it possible to generate more navigation trajectories easily.
>
> Good suggestion! We would like to respond from three aspects:
> 1. We would like to clarify that extending the scale and diversity of SQA3D to construct MSQA involves considerable efforts and contributions, wherein we address a series of challenges including **the design of a more general framework of situated reasoning**, **data quality control**, **efficiency optimization for large-scale generation**, *etc.*
> 2. Feasibility of further scaling up. We agree this could enhance the impact. We think continually scaling up the scenes is theoretically feasible but requires careful consideration and adaptation of different 3D scene assets. For example, ARKitScenes contains fewer multi-view images and shows poorer 3D reconstruction quality (*e.g.*, lower point cloud density). Despite the obstacles, we think continually scaling up 3D scenes is necessary and will stick to it.
> 3. Incorporating synthetic scenes. We agree with the significant potential of synthetic scenes, particularly due to their scalability and the ease of generating annotations. While they can greatly enhance scene diversity, the domain gap between synthetic scenes and real scenes is inevitable, *e.g.*, realism in layout, texture, and lighting. Our current focus on **real-world scans** avoids synthetic noise, but expanding via our pipeline raises the **sim-to-real gap** issue, which will require careful future analysis. We welcome effort on this front. Anyway, incorporating synthetic scenes is still a promising future direction for us.
> ---
> > 4. Missing related work. A seemingly very relevant work on 3D situational awareness [A] is missing in the current manuscript.
>
> Thanks for your reminder. We were pleased to see similar insights in this work. However, since the paper was not yet available on arXiv at the time of our submission, we were unable to cite and discuss it in our paper. We will add a detailed discussion of it in our revised version.
>
> ---
>
> > Addtional Feedback
> > 1. Given the potential disconnect between your current benchmark and real-world embodied AI scenarios, how might you modify your dataset to place more emphasis on visual inputs for situational understanding? Have you considered creating scenarios where the agent must rely primarily on visual cues to determine its situation?
>
> To emphasize more on visual inputs for embodied AI, a plausible solution is to adjust the system prompt and scene-image-text sequence, *e.g.*, reduce textual information and replace object-centric images(or original images) with egoview images to mimic embodied agents. We can render the egoview images in 3D-VL-oriented scenes (*e.g.*, ScanNet) or directly obtain them in embodied-AI-oriented scenes (*e.g.*, Habitat).
>
> ---
>
> > 2. While MSQA offers a large number of 3D scenes, have you explored the possibility of significantly scaling up the dataset, (...), and how might such an expansion enhance the contribution of your work?
> > 3. Your LLM-based dataset curation pipeline seems well-suited for automation. Have you considered (...) What benefits and challenges do you anticipate in such an endeavor?
>
> We indeed expand our dataset to more 3D scene assets such as 3RScan and ARKitScenes. According to our empirical results, data scaling consistently improves the model's performances. We believe extending to more high-quality 3D scene assets can provide abundant resources for the development of powerful situated-reasoning models. The main challenges for scaling up include assets quality, Sim2Real gap, *etc.*, which have been elaborated in our response to Weakness 3.
>
> ---
>
> > 4. The omission of the relevant work on 3D situational awareness [A] is notable. How does your work relate to or differ from this study, and how might incorporating a discussion of this paper enhance your literature review?
>
> We will add a detailed discussion about this work in our revised version. In a nutshell, SIG3D[1] mainly introduce a sophisticated model approach to handling the situated reasoning task. In contrast, we propose a general formulation of this task, a scalable pipeline for automatic data collection, a large-scale dataset for situated QA and navigation, as well as insights into such tasks such as scaling analyses.
>
> [1] Man, Y., Gui, L. Y., Wang, Y. X. Situational Awareness Matters in 3D Vision Language Reasoning. In CVPR 2024.

---

> > ### Comment · Reviewer_pZjR · 2024-08-30
> > **Thanks for the authors' response**
> >
> > I have carefully examined the comments by other reviewers and the response provided by the authors. The response of the authors resolved most of my concerns -- In the rebuttal the authors have agreed and promised to keep scaling the size of the dataset (number of scenes), incorporating synthetic datasets, and adding the discussion of the missing literature. Hence, I have decided to keep my positive rating.

---

### Decision · Program_Chairs · 2024-09-26

**Decision:**

Accept (Poster)

**Comment:**

This paper initially received 677.
As strengths, the reviewers highlighted:
- Potential impact in 3D computer vision and embodied AI
- The interleaved setting of the visual/text
- Manual review to ensure quality

As weaknesses, the main criticisms were:
- The evaluation does not necessarily rely that much on visual input - the questions can lean more towards language based understanding.
- Scale.  However, the dataset already contains 251K reasoning pairs, and authors provide a discussion around the limitations and feasibility of scaling up in the rebuttal.
- Bias by using OpenAI models in-the-loop and in evaluation. The authors addressed this in the rebuttal by also using an Anthropic model, with similar results.

Overall, the only major outstanding criticism is that the questions lean more towards language than visual understanding. The authors do point out that they are testing reasoning, which language can help with. Whist this is a shortcoming and should perhaps be investigated further (e.g. a score on a subset of questions known to require visual reasoning, such as those with the poorest language-only scores), I believe that this issue is outweighed by the thoroughness of the rest of the paper, and the reviews are all positive. I am recommending accept, on the condition the authors include the Anthropic results and discussion around scale.